Li et al. Genome Biology        (2020) 21:203

**RESEARCH**                                                                                          **Open Access**

# The asynchronous establishment of chromatin 3D architecture between in vitro fertilized and uniparental preimplantation pig embryos

Feifei Li[1†], Danyang Wang[1,2†], Ruigao Song[2,3†], Chunwei Cao[3,4†], Zhihua Zhang[1,2*†] [iD], Yu Wang[3,5], Xiaoli Li[1,2], Jiaojiao Huang[3], Qiang Liu[3], Naipeng Hou[3,5], Bingxiang Xu[1,2], Xiao Li[1,2], Xiaomeng Gao[1,2], Yan Jia[1], Jianguo Zhao[2,3] and Yanfang Wang[5*]

* Correspondence: zhangzhihua@
big.ac.cn; wangyanfang@caas.cn
†Feifei Li, Danyang Wang, Ruigao
Song, Chunwei Cao and Zhihua
Zhang contributed equally to this
work.
¹CAS Key Laboratory of Genome
Sciences and Information, Beijing
Institute of Genomics, Chinese
Academy of Sciences, and China
National Center for Bioinformation,
Beijing 100101, China
⁵Institute of Animal Science,
Chinese Academy of Agricultural
Sciences, Beijing 100193, China
Full list of author information is
available at the end of the article

## Abstract

**Background:** Pigs are important animals for agricultural and biomedical research, and improvement is needed for use of the assisted reproductive technologies. Determining underlying mechanisms of epigenetic reprogramming in the early stage of preimplantation embryos derived from in vitro fertilization (IVF), parthenogenesis, and androgenesis will not only contribute to assisted reproductive technologies of pigs but also will shed light into early human development. However, the reprogramming of three-dimensional architecture of chromatin in this process in pigs is poorly understood.

**Results:** We generate three-dimensional chromatin profiles for pig somatic cells, IVF, parthenogenesis, and androgenesis preimplantation embryos. We find that the chromosomes in the pig preimplantation embryos are enriched for superdomains, which are more rare in mice. However, p(s) curves, compartments, and topologically associated domains (TADs) are largely conserved in somatic cells and are gradually established during preimplantation embryogenesis in both mammals. In the uniparental pig embryos, the establishment of chromatin architecture is highly asynchronized at all levels from IVF embryos, and a remarkably strong decompartmentalization is observed during zygotic genome activation (ZGA). Finally, chromosomes originating from oocytes always establish TADs faster than chromosomes originating from sperm, both before and during ZGA.

**Conclusions:** Our data highlight a potential unique 3D chromatin pattern of enriched superdomains in pig preimplantation embryos, an unusual decompartmentalization process during ZGA in the uniparental embryos, and an asynchronized TAD reprogramming between maternal and paternal genomes, implying a severe dysregulation of ZGA in the uniparental embryos in pigs.

**Keywords:** Pig, Early embryos, Chromatin 3D architecture, In vitro fertilization, Parthenogenesis, Androgenesis

## Background

The domestic pig (*Sus scrofa domesticus*) is widely regarded as an important agricultural livestock species and also a crucial biomedical animal model. Given the remarkable similarity to humans, e.g., comparable size and similar early-stage embryonic development [1], the pig provides a unique model for human developmental biology research. Assisted reproductive technologies, including in vitro fertilization (IVF), parthenogenesis (PA), and androgenesis (AG), are widely used in pig reproduction and early-stage embryo development studies. Particularly, the delayed development in PA and AG embryos compared with IVF was overly severe. Thus, understanding the molecular mechanisms utilized by early stage embryos will not only contribute to increased litter size for commercial pig breeding and production, but also to the creation of genetically modified pig models for biomedical research [2]. It is well known that intensive epigenetic reprogramming and chromosome remodeling in preimplantation embryos is a prerequisite to establish a successful pregnancy [2] and that epigenetic reprogramming is reliant on the 3D chromatin architecture of physical bases. However, the three-dimensional structure of chromatin and its reprogramming in preimplantation development remain poorly understood for pig.

The three-dimensional (3D) architecture of the genome plays a central role in the function of nuclei and found drastic reprogramming during early embryonic development [3, 4]. Our understanding of chromatin architecture has been deepened by chromosomal conformation capture-based methods, such as Hi-C and ChIA-PET [5–7]. In eukaryotic nuclei, interphase chromatin is folded in hierarchical structure, including chromosome territories (CT), compartments A and B, topologically associating domains (TADs), and chromatin loops [8–10]. Compartments A and B are associated with open and dense chromatin, respectively. TADs appear to be fundamental structural units, which are conserved across different cell types and across species. Regulatory interactions between promoter and enhancers are believed to have gone through chromatin loops. Structural proteins (e.g., CTCF and cohesin) play central roles in chromatin organization [11]. Biophysics models have been proposed, and a loop extrusion model has drawn much attention in recent years [12]. Genome-wide investigation of chromatin structure in early embryonic development has been possible with the development of low-input and single-cell Hi-C [13, 14], and multiple datasets have appeared in the literature for model systems, such as human [15], mouse [16, 17], zebrafish [18], and *Drosophila* [19, 20], as well as for nonmodel systems, such as medaka [21]. Overall, results obtained for all species studied indicate a reconstruction of chromatin structure. In mouse and *Drosophila*, severely weakened or a lack of structural features (e.g., A/B compartments or TADs) were observed before zygotic genome activation (ZGA), while structural organization emerged at the onset of or immediately after ZGA [16, 17, 19]. In a preprint paper, the authors also reported the emergence of weak structures during ZGA in medaka embryos [21]. In zebrafish, chromatin structure displays a unique systemic loss and regain pattern and the structured organizations are lost when ZGA [18]. In human, TAD is gradually established while compartments are lost in 2-cell embryos and are re-established during embryonic development [15]. The pattern and rate of the chromatin architecture establishment vary from species to species. Thus, species-specific patterns of chromatin dynamics are critical to understand characteristics of embryo development and abnormalities of special in vitro embryos.

In this study, we generated Hi-C maps for somatic cells and various types of preimplantation embryos (IVF, PA, and AG) in pigs. Notably, we found a potential pig-specific feature for the chromosomes in preimplantation embryos. The compartment domains that larger than 10M (i.e., superdomains) were much more prevalent in pig embryos than mouse embryos. Except for the superdomains, the p(s) curves, compartments, and TADs were largely conserved in somatic cells and were gradually established during preimplantation embryogenesis in both mammals. Remarkably, the establishment of chromatin architecture was highly asynchronized during the development of uniparental pig embryos, especially the strong decompartmentalization was seen during zygotic genome activation (ZGA). Our data not only provide valuable resources for evolutionary research on genome architecture, but also can be used as a unique reference for IVF and uniparental pig reproduction.

## Results

### Chromatin architecture is largely conserved in embryonic fibroblast cells between mouse and pig

Using pig embryonic fibroblasts (PEFs), we performed sisHi-C assay and obtained highly reproducible Hi-C maps between two replicates (Table S1 and Fig. S1A), with an average of 256 million *cis*-paired reads for each replicate. The reproducibility of the Hi-C data was validated by HiCRep, GenomeDISCO, and Pearson correlation coefficient (PCC) analyses of contact matrices between replicates (Fig. S1A, B and C) [22]. The Hi-C contact map of PEF showed canonical chromatin organization, including compartments and TAD (Fig. 1a). By comparing to mouse embryonic fibroblasts (MEF), we assessed the similarity of genome architectures between pig and mouse at the chromatin contact frequency curve p(s), A/B compartments, and TAD levels [23] and found that the two mammalian genome architectures were largely conserved. At the p(s) level, the similarity of two p(s) was assessed by Jensen-Shannon divergence (JSD). We found the JSD between PEFs and MEFs were 0.0035, which is comparable to the average JSD of replicates of all our samples (0.0038) (Fig. 1b). At the level of compartment, the two cell types showed similar arrangement of compartment assignments. There were 72.6% of homologous bins (50 kb) in PEFs that had identical compartment assignments in MEFs (Fig. 1c, see Materials and Methods), and compartment strengths between PEFs and MEFs were also comparable ($p = 0.373$, Fig. S1D). The accuracy of compartment assignments in PEFs were validated by our ChIP-seq data of two histone modification marks, H3K27ac and H3K4me3 (Table S1), and a public RNA-seq data [24]. The A compartment showed higher levels of gene expression ($p = 1.84E–7$, Fig. 1d) and also showed an enrichment of H3K4me3 and H3K27ac modification marks ($p = 0.001$, Fig. 1e), compared with B compartments. At the TAD level, these two mammals also showed remarkable conservation. There were 6038 TADs identified with a median size of 360 kb in PEFs using our recently developed algorithm deDoc [25]. The accuracy of TAD identification was evidenced by the aggregate analysis of insulation score (Fig. 1f), and further supported by the enrichment of TSS, and CTCF binding in the TAD boundaries (Fig. 1g). Moreover, CTCF motifs within the TAD boundaries showed a clear convergent distribution (Fig. 1h), suggesting that loop extrusion may also work in pigs [8]. We found that 33.88% of pig homologous boundaries were shared in mouse

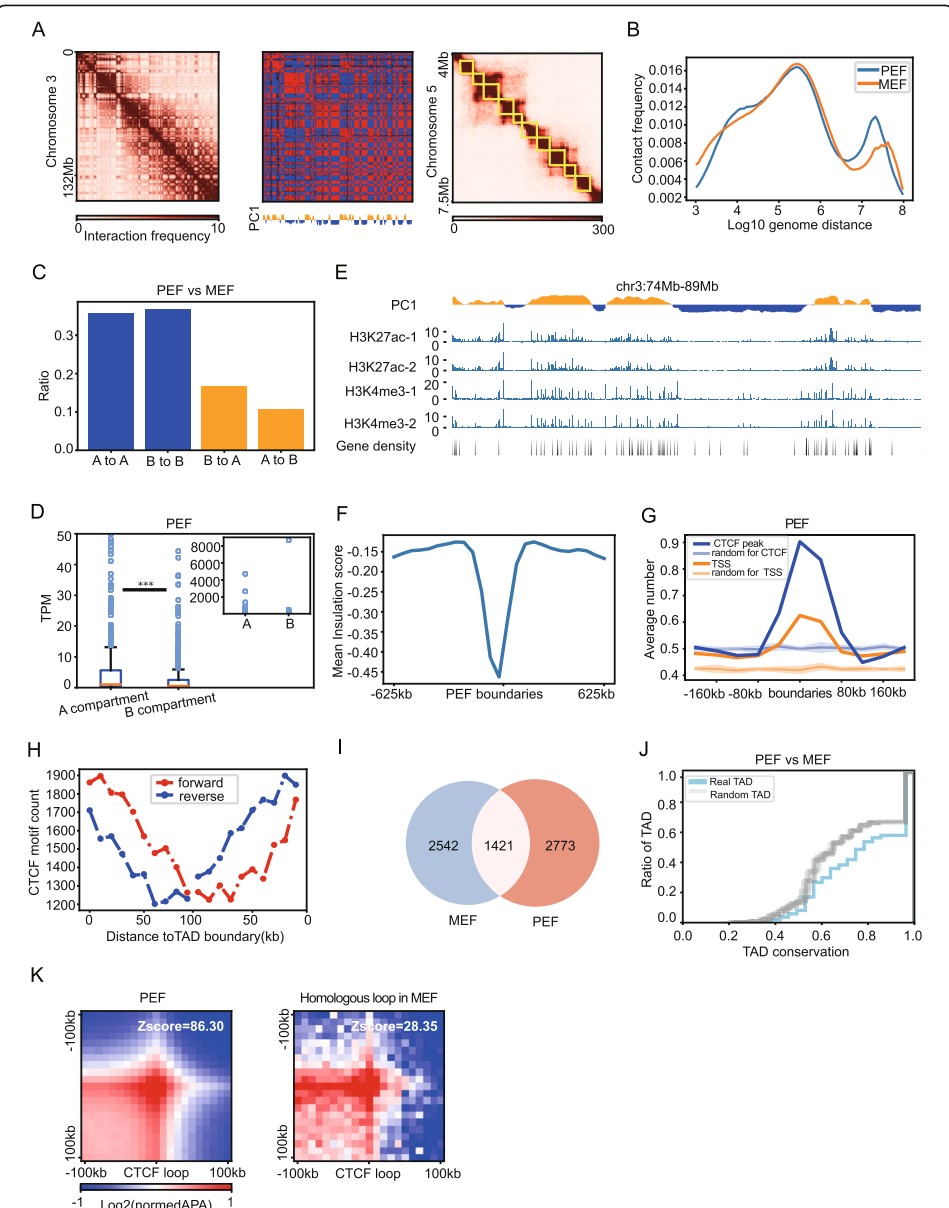

**Fig. 1** Chromatin architecture of pig embryonic fibroblast. **a** Left: Hi-C contact matrix of chromosome 3 at 100 kb resolution at PEF (pooled data from 2 biological replicates). Middle: the principal component 1 (PC1) values (bottom) and correlation matrix (upper) obtained at 300 kb resolution for chromosome 3. Positive PC1 values represent compartment A regions (yellow), and negative values represent compartment B regions (blue). Right: TADs identified by deDoc are shown (50-kb resolution, chr5: 4–7.5 Mb). Yellow boxes represent TADs. **b** Relative contact frequency plot showing the percentage of contacts as a function of genome distance in PEF (blue) and MEF (orange). **c** Ratio of homologous bins classified by A/B compartment status between PEF and MEF. **d** Comparison of expression level between genes located in A compartments and genes located in B compartments of PEF ($P = 1.84\text{E}{-}7$, Mann-Whitney $U$ test, one-tailed). **e** Plots showing H3k27ac and H3k4me3 ChIP-seq signal relative to the compartments (chr3: 14–19 Mb). Gene density is also shown. **f** Mean value of insulation scores around PEF TAD boundaries (boundaries ± 625 kb). **g** Average numbers of CTCF ChIP-seq peaks (blue) and gene TSS (orange) around PEF TAD boundaries (boundaries ± 200 kb). **h** Motif count and orientation of inferred CTCF binding sites relative to PEF TAD boundaries. **i** Venn graphs showing the overlap of TAD boundaries between PEF and MEF ($P = 7.63\text{E}{-}67$ compared to random, Fisher's exact test). **j** Cumulative curve of TADs according to TAD conservation. TAD conservation was calculated as the probability of two homologous bins located within a TAD of PEF which also co-existence within a TAD of MEF. The same curve under shuffled TADs was shown in gray as control. **k** Aggregate loop plots showing the strength of PEF CTCF loop (left) and strength of interactions between homologous loop anchors in MEF (right)

($p$ = 7.63E–67, Fisher exact test, Fig. 1i), which is even higher than the conservation between mouse and human (18.05% comparing MEF and IMR90, Fig. S1E). In addition, we calculated TAD conservation between MEFs and PEFs as the fraction of homologous bin pairs within a TAD in pig that is also located within one TAD in mouse. Conservation between PEFs and MEFs is about 79.29%, which is significantly higher than controls using shuffled TAD (72.93%, $p$ = 2.49E–56 rank-sum test, Fig. 1j and Fig. S1F). At the chromatin loop level, a total of 4489 CTCF loops were detected for PEFs at 10 kb resolution (Materials and Methods). Aggregate Peak Analysis (APA) analysis showed the enrichment of interaction between the homologous loop anchors in MEF (Fig. 1k). Together, our data showed that the pig genome is organized in compartments, TADs, and loops, and the overall architecture of chromatin is conserved between PEFs and MEFs.

### The prevalence of superdomains in pig preimplantation embryos

To address whether the pig and mouse shared similar patterns in chromatin architecture reprogramming in the preimplantation embryos, we collected pig IVF, PA, and AG preimplantation embryos at the zygote, 4-cell, and morula stage, and all embryonic stages were characterized using Hoechst staining (Fig. 2a). We performed sisHi-C assay and an average of 56 million *cis*-paired reads were generated for each stage (Fig. 2b, Table S1). The reproducibility of Hi-C data was validated using HiCRep, Genome-DISCO, and PCC between replicates (Fig. S1A, B and G) [22]. Here, we will discuss the dynamics of IVF embryos and later compare to PA and AG embryos. To compare developmental stages, we selected to consider zygotes, before ZGA and after ZGA stages in pig and mouse preimplantation embryos. As the major ZGA occurs in 2-cell and 4-cell stages in mouse and pig, respectively, we compared zygotes, early 2-cell, and 8-cell mouse embryos with the zygotes, 4-cell, and morula pig embryos, respectively. The 8-cell mouse embryo and pig morula embryo were comparable as they experienced similar numbers of cell cycles after ZGA [26]. To make the Hi-C data of different samples comparable, we downsampled the pooled valid-read pairs of each sample to the number of AG 4-cell embryos that had the lowest valid contacts.

Intriguingly, by visual inspection of the PC1 of Hi-C matrices in pig preimplantation embryos (Fig. S2A), we noticed that many chromosomes were made up of only a few superdomains, where the compartment domains were larger than 10 Mb. For example, three superdomains almost covered the whole chr6 (Fig. S2A). For chr3, chr4, chr8, and chr10, the chromosome arms could even be clearly separated by the PC1 (Fig. S2A). However, such superdomains were much less prevalent in mouse preimplantation embryos. The superdomains (> 10 MB) covered 54.88%, 59.33%, and 47.70% of the genome regions in zygote, 4-cells, and morula pig embryos, respectively, while only 28.22% ($p$ = 7.95E–271), 30.69% ($P$ = 1.17E–308), and 24.11% ($p$ = 1.32E–222) of the genome regions were covered by superdomains in zygotes, early 2-cell, and 8-cell mouse embryos, respectively. To quantify the prevalence of those superdomains, we plotted the accumulative curve of genome coverage as a function of domain size (Fig. 2c). The larger domains covered many more genome regions in pig preimplantation embryos than those in mouse (Fig. 2c). These superdomains in pig embryo were not caused by the sparseness of the downsampled matrices, as the identical pattern can also be seen in the raw data (Fig. S2B). Thus, the enrichment of superdomains might

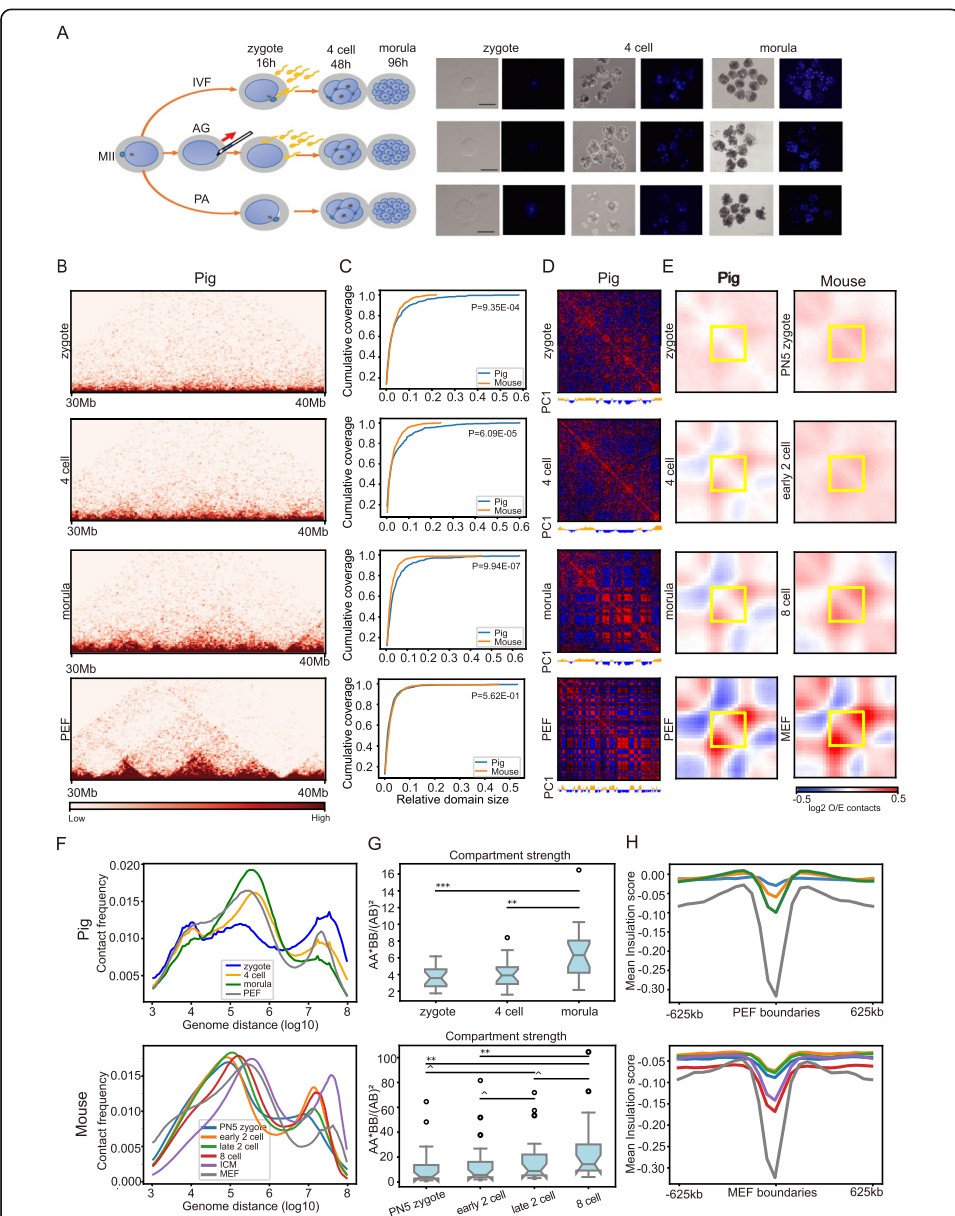

**Fig. 2** Reprogramming of chromatin structure during pig in vitro fertilized early embryogenesis. **a** Schematic of generation of IVF, PA, and AG embryos (left) and microscopy images of 1-cell, 4-cell, and morula embryos staining with Hoechst (right). **b** Normalized Hi-C contact heatmap of 10 Mb region in chromosome 3 were shown at 50 kb resolution in the three stages of IVF embryos and PEF. **c** The accumulative curve for genome coverage by domains size. X-axis represents the relative domain size, i.e., domain length/chromosome length. Only domains larger than 1 MB were included in this figure. The P values are given for the t test of average domain sizes. **d** PC1 and correlation matrix of chr14 at 100 kb resolution. **e** Observed/expected (O/E) aggregate plot of TADs in pig and mouse. **f** The contact frequency decay curves of Hi-C data during pig (upper) and mouse (bottom) embryogenesis. **g** Compartment strength which were calculated as $AA*BB/AB^2$ for each chromosome in embryos of pig (upper) and mouse (bottom). Compartments identified in morula and 8-cell stages were used as a reference for pig and mouse, respectively. **h** Mean value of insulation scores around PEF TAD boundaries (upper) and MEF TAD boundaries (bottom). The color code is identical to **f**. $*P < 0.05$, $**P < 0.01$, $^{\wedge}P < 0.1$

be a pig-specific feature of preimplantation embryos: however, further investigation is needed to validate this speculation.

### Gradual establishment of chromatin architectures during pig IVF early embryogenesis

In general, chromatin architecture was gradually established during IVF early embryogenesis (Fig. 2b). During the process, distant contacts (> 10 M) gradually decrease, intermediate distant contacts (> 50 k and < 1 M) gradually increase, and local contacts (shorter than about 20k) gradually decrease as shown by p(s) curves (Fig. 2f). Quantitatively, the JSD of p(s) curves between morula and zygotes, and between morula and 4-cell embryos are 0.0227 and 0.0044, respectively. The decreased long distant contact may indicate a gradual decompaction of chromatin, while the increased intermediate distant contacts may indicate the establishment of a megabase-sized modular structure, such as TADs (Fig. 2e). Unlike the gradual decompaction of chromatin in pig, i.e., decreased distal contact frequencies, the distal contact gradually increased in mouse during the development process (Fig. 2f).

Chromosomal segregation levels are gradually increased during pig embryogenesis, especially after ZGA (4-cell embryo, Fig. 2d and Fig. S2A). To quantify the segregation level, we identified compartments in IVF morula and took these as a reference to calculate the compartment strength ($AA^*BB/AB^2$) of other developmental stages. We found that compartment strength gradually increased during development, especially from 4-cell stage and morula, which showed significant enhancement (Fig. 2g). The gradually increasing segregation level could also be seen when using saddle plots to quantify compartment strength (Fig. S3A). The gradual compartmentalization could also be observed in mouse embryogenesis with a significant enhancement after ZGA (2-cell of mouse) as well (Fig. 2g and S3B). Furthermore, we found that the segregation level of the chromosomes in pig morula and mouse 8-cell embryos did not reach that in PEF and ICM, respectively (Fig. 2d), suggesting that the compartments have not be fully established at these stages [16].

At the TAD level, we found TADs were also gradually established in pig's preimplantation embryos (Fig. S4A). To quantitatively assess the establishment of TADs, we calculated the aggregate TAD signal by using the TADs identified in PEFs as the reference. Indeed, aggregate TAD signals are weak in zygotes (average = 0.530) and gradually increase during development (0.551 and 0.563 in 4-cell stage and morula, respectively, Fig. 2e). The gradual enhancement of TADs can also be seen with decreasing insulation score (IS) at the TAD boundaries (– 0.0295, – 0.0596, and – 0.0995 for the zygote, 4-cell, and morula stage, respectively, Fig. 2h). Furthermore, the overall arrangement of TADs gradually approaches the patterns seen in PEFs. Using our deDoc algorithm, we de novo identified TADs at the zygotes, 4-cell, and morula stage [25] and revealed 3822, 3347, and 3906 TADs, respectively. The TADs in PEFs were substantially similar to those in morula and less similar to the TADs in the 4-cell and zygote stage (Fig. S4B). However, both aggregate TAD signal and IS at the morula stage were still substantially weaker than those in PEFs (0.654 and – 0.317, respectively). Thus, the TAD structures are gradually established, but not fully established until the morula stage. Finally, for the TAD, the aggregate TAD signals also suggest a gradual establishment in zygotes, before and after ZGA stages in mouse (0.573, 0.563, and 0.593, respectively, Fig. 2e). This pattern can also be observed with the IS (Fig. 2h). Thus, the pattern of compartmentalization and TAD establishment

is similar during pig and mouse embryogenesis when aligning the developmental stages based on ZGA.

At the chromatin loop level, we found that a small fraction of CTCF chromatin loops in PEFs may be established in pig preimplantation embryos, yet few such loops in MEFs are established before the ICM stage in mouse (Fig. S5, Supplementary Text).

### The overall dynamics of chromatin architecture in embryos from PA and AG are distinguished from IVF

The uniparental duplications, such as PA and AG in mammals, resulted in severe embryonic defects or lethality due to genetic asymmetries between parental genome [2, 27]; however, they provided a unique model to investigate the dynamics of uniparental chromosomes during the development of preimplantation embryos. Unlike the gradual development process of IVF preimplantation embryos, both PA and AG embryos experience substantial global chromatin architecture alterations. A clear trend can be seen in the Hi-C heatmap where the majority of the Hi-C reads are concentrated in the narrow range near the diagonal with both of replicates showed similar (Fig. 3a, Fig. S6). This trend is also shown in the p(s) analysis (Fig. 3b), where remarkably higher peaks of local contact (shorter than about 20k) are observed in 4-cell and morula stages of PA and AG and in zygotes of PA preimplantation embryos compared with IVF embryos. Intriguingly, no local contact peaks are seen in the mouse allelic p(s) curves (Fig. 3c). In addition to the local contact, the progress of distal (> 10 M) and intermediate contacts (< 1 M and > 50 K) are also significantly different in more than half of the samples when comparing uniparental embryos to IVF embryos (Table S2). In PA embryos, the distal contacts are depleted in zygotes while they are drastically accumulated at the ZGA stage (4-cells), finally dropping back to mid-level at morula stages, which is equivalent to that seen at IVF morula stages (Fig. 3b). The intermediate contacts are enriched in zygotes, depleted at the 4-cell stage, and accumulated again at the morula stage. In AG embryos, however, the developmental trajectory is the opposite of that in PA embryos for the distal contact frequency, which first decreases at the 4-cell stage and then accumulates at the morula stage, whereas the trajectory for intermediate contacts is similar to that in PA. These drastic alterations of contact frequencies have not been seen in mouse allelic chromatin (Fig. 3c); thus, these differences in PA and AG chromatin architecture may reflect the abnormal nature of those uniparental embryos.

We wondered if these potentially abnormal dynamics of chromatin architecture during the development of PA and AG preimplantation embryos could be associated with the deficient inter-chromosomal contacts between the homologous chromosomes, especially since a considerable portion of collected uniparental embryos might be in fact haploid [27]. To examine this possibility, we performed a meta-analysis on inter-homologous-chromosomal Hi-C reads to calculate the frequency of the homologous contacts during mouse embryogenesis [16]. We found limited homologous contacts in zygotes; however, the number of contacts sharply increased from the zygote to the late 2-cell stage and finally to the 8-cell stage, while they slightly decreased at the ICM stage (Fig. 3d). Since the chromatin architecture also emerged at the onset of ZGA in mouse, we wondered whether these homologous interactions are associated with the establishment of chromatin spatial features. To determine this possibility, we plotted the accumulative distribution

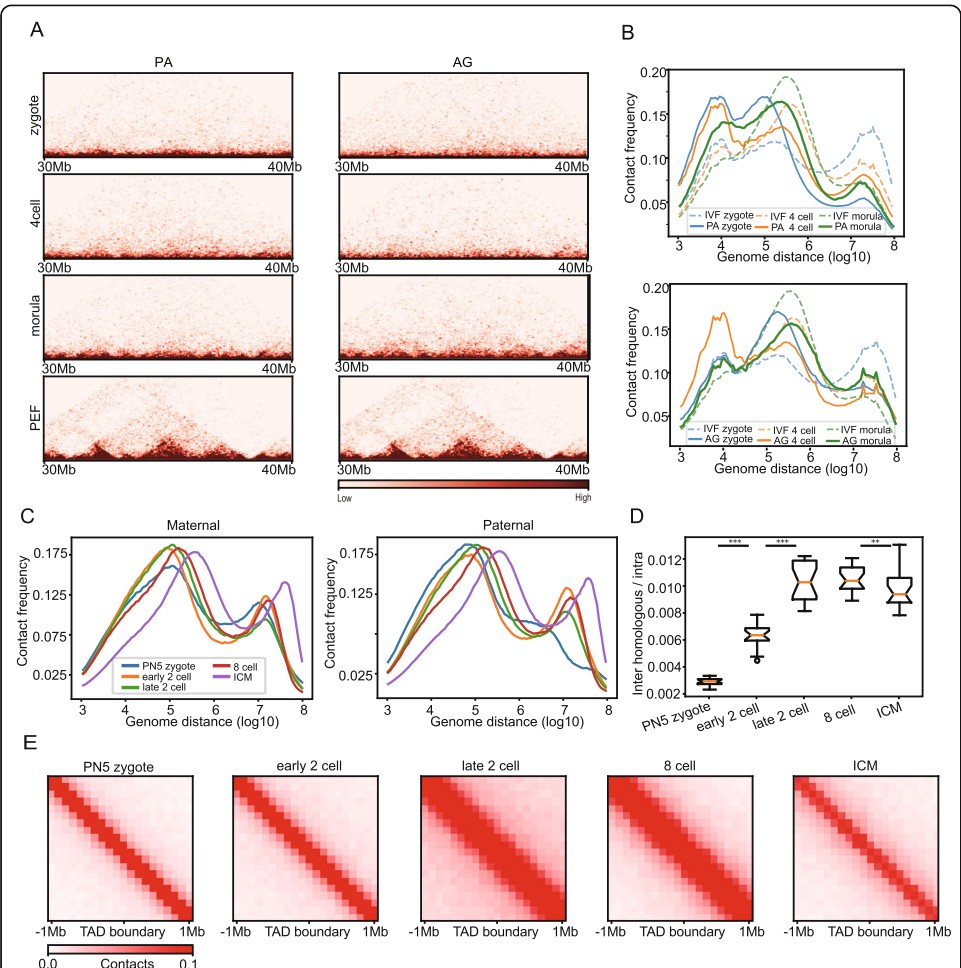

**Fig. 3** The overall dynamics of chromatin architecture in embryos from PA and AG are distinguished from IVF. **a** Normalized Hi-C contact heatmaps for the same 10 Mb region as Fig. 2b were shown in three stages of PA (left) and AG embryos (right). **b** The contact frequency decay curves of pig Hi-C data during PA (upper) and AG (bottom) embryogenesis. The curves for IVF embryos are also shown. **c** The contact frequency decay curves of mouse Hi-C data in maternal (left) and paternal (right) alleles. **d** Ratio of contacts between inter-homologous-chromosome and contacts of intra-chromosome for each chromosome during mouse embryogenesis. **e** Aggregate plot showing the distribution of inter-homologous-chromosome interactions (100-kb resolution) at MEF TAD boundaries and nearby regions (boundaries ± 1 Mb) during mouse embryogenesis. \*\*\*$P < 0.001$, \*\*$P < 0.01$

of the homologous contacts flanking the TAD boundaries. Our rational was that if the homologous contacts contribute to the establishment of chromatin architecture, it should not be evenly distributed along the chromosomes but be concentrated at certain structure features. However, these interactions were observed to be evenly distributed along the genome and were not enriched at TAD boundaries or within TADs (Fig. 3e). This data suggests that communication between the homologous chromosomes may not be directly associated with the establishment of TADs.

### The development of A/B compartments in PA and AG preimplantation embryos is unique at the 4-cell stage

We compared the compartmentalization progress between PA, AG and IVF preimplantation embryos in pig and found divergence between the uniparental and IVF embryos

(Fig. 4a and Fig. S7A). Although the compartmentalization pattern was similar between PA and AG embryos (Fig. S7B), the strength of compartmentalization were different. Using the compartments identified in IVF morula as the reference, we calculated compartment strengths (AA*BB/AB²) in all samples. Visually, compartmentalization was found to be stronger in AG and PA zygotes than that in IVF zygotes (Fig. 4a and 2c). Quantitatively, the compartment strengths of AG zygotes are significantly higher than those of IVF zygotes, while PA zygotes are moderately higher than IVF zygotes (Fig. 4b). In the saddle plots, both PA and AG zygotes also showed higher segregation levels than IVF zygotes (Figs. S7C, D and S3A). However, the A/B compartment arrangements were similar between the three types of embryos at the zygote stage. PCC of PC1 between uniparental and IVF embryos at the zygote stage were shown to be 0.709 and 0.740 for PA and AG, respectively (Fig. 4c and Fig. S8). Strikingly, unlike the gradual compartmentalization in IVF embryos, we noticed a decompartmentalization process in both PA and AG embryos at the 4-cell stage (Fig. 4a and Fig. S9 for all chromosomes). The average compartment strengths were 3.566 and 2.769 in the PA and AG 4-cell stages, respectively, which were significantly lower than those in corresponding zygotes ($p = 0.065$ and 0.0006, respectively) (Fig. 4b). In addition, this decompartmentalization process resulted in a substantial alteration of A/B compartment assignment over the whole genome, as evidenced by the drastic decrease in the PCC of the PC1 between uniparental and IVF embryos at the 4-cell stages (PCC = 0.620 and 0.558 for PA and AG, respectively), as well as between PA and AG embryos (PCC = 0.563, Fig. 4c and Fig. S8). At the morula stage, the compartmentalization was partially reestablished in

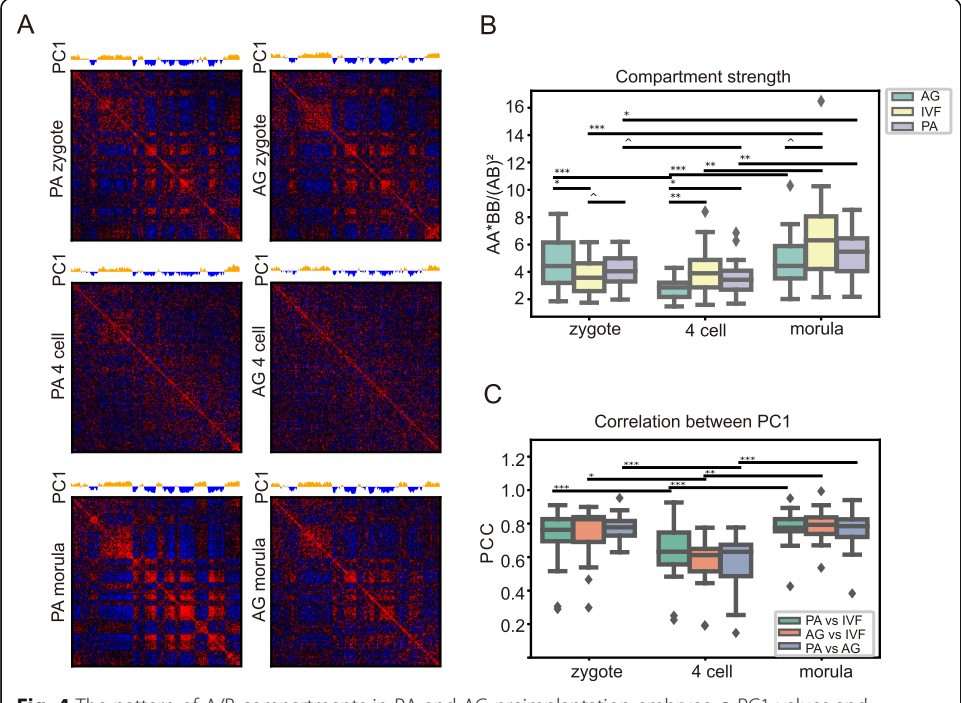

**Fig. 4** The pattern of A/B compartments in PA and AG preimplantation embryos. **a** PC1 values and correlation matrix of chromosome 14 in PA (left) and AG (right) preimplantation embryos. **b** Compartment strength in IVF, PA, and AG preimplantation embryos. **c** Boxplots show Pearson correlation coefficient of the PC1 between the three kinds of reproductive embryos for each developmental stage. ***$P < 0.001$, **$P < 0.01$, *$P < 0.05$, ^$P < 0.1$

PA and AG embryos, but the degree of compartment establishment in AG was weaker than PA, while the degree was similar between PA and IVF morula (Fig. 4a and 2c). Weaker compartmentalization in AG morula was also evidenced by the average compartment strengths, showing 6.589, 5.306, and 4.934 for IVF, PA, and AG morula, respectively ($p = 0.055$ comparing AG morula with IVF morula, Fig. 4b). Furthermore, the partial reestablishment make the compartment strength of AG morula is comparable with AG zygote ($p = 0.385$). Moreover, at the morula stage, compartment reestablishment in uniparental embryos resulted in a higher similarity of PC1 with IVF morula (Fig. 4c and Fig. S8). The decompartmentalization of PA and AG at the 4-cell stage and the relatively weaker compartmentalization of AG morula can also been seen in the saddle plots (Fig. S7C and D). Together, these data indicate that the compartmental structure degrades at 4-cell stages and is reestablished at morula stages for both PA and AG preimplantation embryos, but the reestablishment is weaker in AG morula.

### TADs are asynchronously established in PA and AG embryos

Because TAD structures were gradually established in IVF preimplantation embryos and gradually resembled TADs pattern of PEFs, we wondered if this might also be the case for uniparental embryos. Indeed, there was a clear trend of gradually establishing TAD boundaries in both PA and AG embryos when using PEF TAD as a reference (Fig. 5a, b and S10A). This gradual establishment of TAD boundaries was also characterized in de novo identified TADs using our deDoc algorithm (Fig. 5c). However, TAD establishment was slower in AG than in PA, as the absolute ISs of the TAD boundaries were always smaller in AG than in PA at each stage (Fig. 5a, b and S10A). Between the AG and IVF, the absolute ISs were almost identical at the zygote stage, whereas they were much smaller in AG than in IVF at the 4-cell and morula stages (Fig. 5b). Thus, TADs are gradually established and the process is much slower in preimplantation AG embryos compared with PA embryos.

Next, we asked if the asynchronous establishment of TADs in PA and AG preimplantation embryos could also been seen in the paternal and maternal chromosomes in mouse. Similar to uniparental pig embryos, the progress of TAD establishment is faster in mouse maternal chromosomes than those in paternal chromosomes at zygote, early 2-cell, and late 2-cell stages (Fig. 5d). Unexpectedly, the average IS at TAD boundaries for the maternal 8-cell stage was substantially reduced compared to the late 2-cell stage. Nevertheless, even at the ICM stage, considerable divergence in the TAD arrangement remains between the two homologous chromosomes (Fig. S10B), although the TAD strengths are almost equal between maternal and paternal TAD boundaries. Particularly, for chromosomes 7, 14, and X, TADs show rather unique patterns (Fig. S10B). These chromosome-specific, slowly established TAD structures were not seen in the uniparental preimplantation pig embryos (Fig. 5c). Together, our data indicates that TAD establishment gradually develops in pig preimplantation embryos, not only for IVF but also for uniparental embryos. The chromosomes that originated from oocytes show a faster rate of TAD development than chromosomes that originated from sperm.

Last, we identified 303, 277, and 270 boundaries with significantly different IS in zygotes, 4-cell, and morula stages for PA embryos, respectively (Fig. S11A). The encompassed genes were found to be enriched in embryonic development-related GO terms,

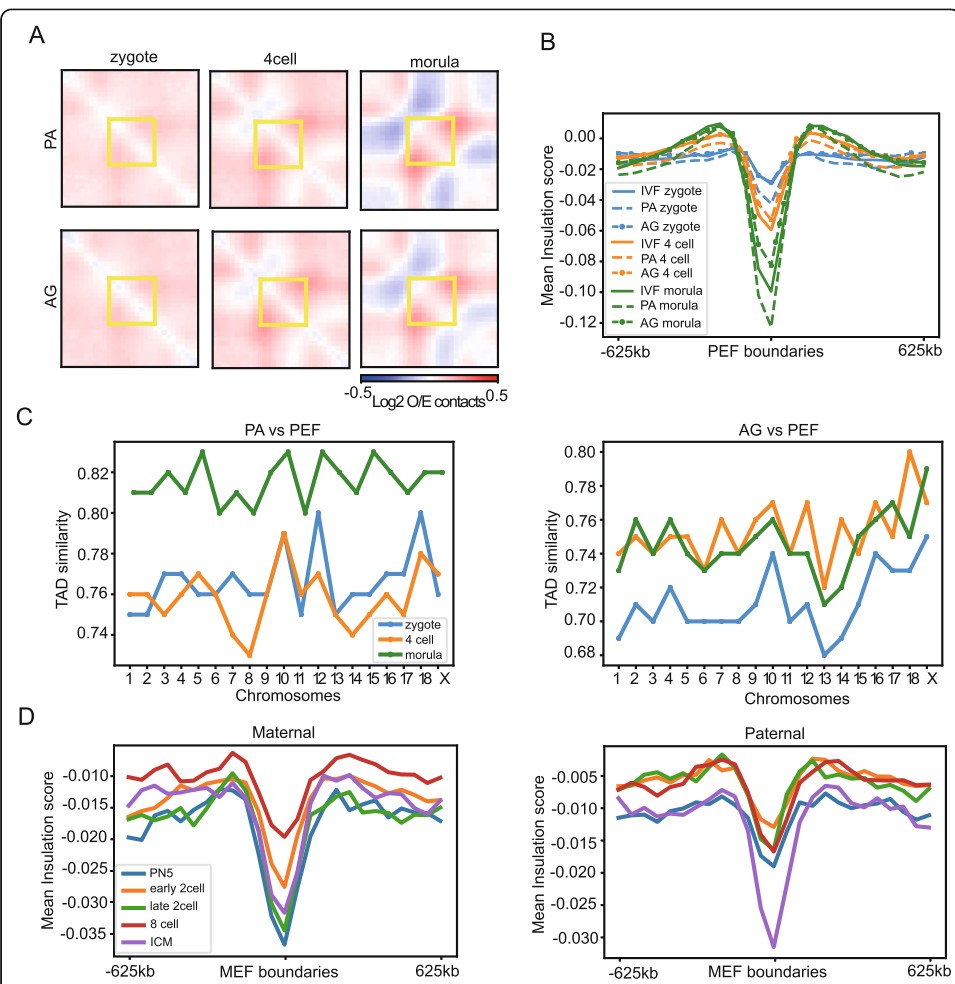

**Fig. 5** The pattern of TADs in PA and AG preimplantation embryos. **a** Observed/expected (O/E) aggregate plot of TADs in PA (upper) and AG (bottom) embryos. **b** Mean value of insulation score at PEF boundaries and nearby regions (boundaries ± 625 kb) in different developmental stages. **c** TAD similarity between PEF and developmental embryos of PA (left) and AG (right) for each chromosome. **d** Mean value of insulation score at MEF boundaries and nearby regions (boundaries ± 625 kb) in mouse maternal (left) and paternal (right) alleles

such as "embryonic organ morphogenesis" and "sperm-egg recognition" (Fig. S11B). For AG embryos, 248, 288, and 248 different boundaries were identified for zygotes, 4-cell, and morula stages, respectively. And the encompassed genes were enriched in pathways associated with embryonic development and response to stimulus (Fig. S11C). Together, our results suggest that the improper establishment of chromosome architecture may be associated with abnormal activation of genes in embryogenesis, which may partially explain the decline of the in vivo developmental potential of pig uniparental embryos.

## Discussion

Here, we present the first high-resolution chromosome conformation profile of pig nuclei and the dynamics of chromatin reprogramming during IVF, PA, and AG embryogenesis. The first notable finding is abnormal development at nearly all hierarchical levels of chromatin architecture in the uniparental pig embryos (Figs. 3, 4 and 5). These

observations are striking for two reasons. First, a remarkably similar pattern of compartmentalization was found in PA and AG zygotes in the current study. In mouse, the paternal and maternal compartmentalization progress has been shown to proceed through a clear distinguished pattern and the allelic differences can be found as late as the 8-cell embryo stage [16]. For example, in the paternal genome, compartmentalization is weakly observed as early as the zygote stage; however, compartmentalization is barely observed in the maternal genome until ZGA in the late 2-cell stage [16, 17]. Thus, the nearly identical establishment of compartments in both PA and AG zygote embryos may imply a unique feature of pig embryo development. Particularly, the early establishment of compartments in PA zygotes should be further investigated. Second, there is an unexpected decompartmentalization process observed during ZGA at the 4-cell stage in uniparental embryos, whereas in mouse, the compartments are mainly established during ZGA [16]. Thus, these results may indicate that the compartmentalization during ZGA is critical to the reprogramming of chromatin. One possible explanation for the abnormality is that uniparental embryos contain a mixture of haploid cells as previously reported [27]. However, the abnormality may also exist in diploid embryos as haploid cells were only a minor percentage of total cells at the morula stage for both PA and AG [27]. Thus, whether this abnormality might relate to the embryonic degeneration observed in uniparental embryos is an interesting question that needs to be further investigated.

The second notable finding is the prevalence of superdomains in pig embryos and the lack in mouse, making superdomains potentially a pig-specific feature. Although further investigation is needed to validate this speculation, we would like to discuss some interesting points here. As the definition of superdomain was based on the distribution of the PC1 eigenvector, the prevalence of superdomain might represent a more outstretched arrangement of the chromosomes. It would be rather interesting to ask what caused this outstretched arrangement and determine if some pig-specific embryonic feature might contribute to this phenomenon. For example, could this phenomenon be a result of the greater number of lipid droplets present in the pig oocytes and embryos than those of other mammalian species [28]? Although the majority of lipid droplets were present in cytoplasm, there were reports that nuclear lipid droplets exist [29]. Important roles for fatty acids and lipid metabolism in the promotion of oocyte maturation and embryo development have been reported [30]. Whether the high lipid concentration plays a role in this phenomenon could be an interesting question to answer.

The third notable finding is the high concentration of local contacts of chromatin structure in pig preimplantation embryos (Figs. 2f and 3b), which were not enriched in mouse (Fig. 3c). This is not likely to be an artifact in the Hi-C library or an error in data processing because the peak corresponding to local contacts was much weaker in PEFs. Thus, we imagine that there might be different compositions of nanoscale chromosomal fiber (10–30 nm structures [31, 32]) within pig and mouse preimplantation embryos. Obviously, this imaginative picture needs further investigation. Together, our data show that the pig genome architecture and its development in preimplantation embryos are noticeably unique.

The main motivation behind our present study is to probe a possible abnormality in chromatin architecture that could be connected to the developmental competence of pig uniparental embryos. We showed several lines of evidence to support the existence

of such a connection. Although the chromatin structure in these uniparental embryos followed an estranged trajectory from the gradual chromatin architecture establishment process observed in IVF embryos, current data may have underestimated the abnormality of chromatin architecture in uniparental embryos and overestimated the normality in the IVF embryos. This might be accounted for by the embryos we selected to sample; our embryos are at least partially successful reprogrammed in their precursive stages and severe abnormalities carried by fatal embryos may not be present in our samples. On the other hand, although we assumed that IVF embryos were normal and could be used as a reference, the effects of in vitro fertilization and in vitro culture conditions are not negligible [33]. Further, the asynchrony we observed between the uniparental embryos did not directly indicate the existences of the asynchrony in vivo between maternal and paternal pig chromosomes. It would be invaluable if homozygote pig lines could be established in the future to address this issue. Finally, we do not have enough data or clues to even attempt to discuss what causal factors might lead to chromatin abnormal development. The early events that trigger the reprogramming of chromatin are still one of the core questions in embryo development that remains unexplored. Transcriptome [34, 35], epigenome [36], and chromatin [16, 17] have all been profiled in mouse preimplantation embryos and have provided events that were found to be intriguingly active at early stages, such as the activity of transposable elements [37]. However, corresponding data remains urgently needed for pig preimplantation embryos. Besides the multiple types of "omic" data, more interpretive experiments are crucial to provide causal relationships between the abnormalities we report here or other aberrations, which have yet to be identified, that contribute to the high failure rate of pig embryos.

## Conclusions

We have shown the 3D chromatin structure of pig nuclei and the reprogramming of chromatin structure during embryogenesis of IVF-, PA-, and AG-derived pig embryos. Our data not only provide a valuable resource for 3D genome evolution research, but also a unique reference for IVF and uniparental reproduction studies. Our findings highlight that proper rate in reprogramming of chromatin architecture may play critical role in the successful development of embryos. Future studies are needed to identify the mechanistic drivers that underlie the abnormal kinetics of chromatin assembly and the establishment of 3D chromatin architecture in the early development of uniparental embryos.

## Methods

### Collection of sperm, oocytes, and PA/IVF/AG early embryos

Porcine follicular oocytes of Large White pigs were collected from follicles (3–6 mm in diameter) in the ovaries, which were collected from prepubertal gilts at a local slaughter house and transported to our laboratory at ~ 38.5 °C in saline. The cumulus-oocyte complexes (COCs) were washed three times in Hepes-buffered Tyrode medium [38] and then cultured for 42–44 h at 38.5 °C under 5% $CO_2$ in air. The COCs were vortexed in 0.1% hyaluronidase in Hepes-buffered Tyrode medium for 4 min to eliminate the cumulus cell contaminants. Then, matured oocytes with an extruded first polar body were selected. Metaphase II (MII) stage oocytes were collected after removal of the first polar body by pronase.

Prior to parthenogenetic activation, the matured oocytes were equilibrated for 30 s at 38.5 °C in an activation medium; the oocytes were activated with two direct current pulses at 1.2 kV/cm for 30 μs provided by a BTX Electro-cell Manipulator 200 (BTX) in activation medium. Then, the activated oocytes were transferred into 500 μL porcine zygote medium-3 (PZM3) and cultured at 38.5 °C in humidified 5% $CO_2$. Zygotes (PN4) were collected 16 h after electrical stimulation, 4-cell embryos were collected 48 h after electrical stimulation, and the morula were collected 120 h after electrical stimulation. Both first and second polar bodies from embryos were removed by pronase.

In vitro fertilization (IVF) was performed as previously described [39]. Briefly, 30 MII oocytes were transferred into 50 μL droplets of modified fertilization Tris-buffered medium (mTBM) containing 10 mM Tris, 2 mg/ml BSA, and 2 mM caffeine. Frozen porcine semen in straws were thawed in 50 °C water for 16 s, then washed three times by centrifugation (1900*g*, 4 min) with sperm wash buffer (DPBS with 1 mg/ml BSA (pH 7.3)). The sperm were incubated in mTBM for 20 min at 38.5 °C in 5% $CO_2$ humidified air. The supernatant with swimming sperm was transferred to a new well. Sperm activity was assessed in a cell counting plate before using and collecting. Fifty microliters of sperm mTBM solution was added to the fertilization droplets. The final sperm concentration was $5 \times 10^5$ cells/ml. MII oocytes were co-incubated with sperm for 5 h. After fertilization, oocytes were transferred into 500 μL PZM3 medium and cultured in four-well dishes at 38.5 °C in 5% $CO_2$ humidified air. Zygotes (PN4), 4-cell embryos, and morula were collected 18.5 h, 48 h, and 120 h after adding the sperm suspension to the fertilization droplets. All the embryos were treated with pronase to remove polar bodies, zona pellucida, and excess sperm. Finally, samples were collected from embryos, which had been confirmed by staining with 1 μg/mL Hoechst 33342 (Sigma).

To obtain AG embryos, the porcine oocyte enucleation and IVF was carried out as previously described [40]. Briefly, MII oocytes were enucleated through gentle aspiration of the first polar body and adjacent cytoplasm, containing chromosomes, with a beveled pipette (17–20 μm) in manipulation medium supplemented with 7.5 mg/ml cytochalasin B. Enclosed cytoplasts were transferred to mTBM until IVF. After being stained with Hoechst, the presence of the maternal genome was examined under a fluorescence microscope. Completely enucleated oocytes were selected and then subjected to IVF using the above protocol. The collection procedure of the zygotes (PN4), 4-cell embryos, and morula of the AG embryos was performed as described for the IVF embryos.

## PEF culture and collection
Porcine embryo fibroblast (PEF) were isolated from 35-day-old fetuses of Large White pigs, which was mechanically dissociated and digested in DMEM supplemented with 0.5 mg/mL collagenase IV for 2.5 h at 38 °C. Isolated PFFs were cultured in the Dulbecco's modified Eagle's medium (DMEM, HyClone) supplemented with 15% fetal bovine serum (FBS, Gibco), 2 mM GlutaMAX (Gibco), and 1% nonessential amino acids (NEAA, Gibco) for 24 h. Animal experiments were performed according to the guidelines of the Experimental Animal Welfare and Ethical of the Institute of Animal Science, Chinese Academy of Agricultural Sciences (IAS2018-11).

### Preparation of sisHi-C libraries

The sisHi-C was conducted according to previously described protocols [16, 17]. Briefly, samples were fixed with a final concentration of 1% formaldehyde and quenched with 0.125 M glycine. Cells were lysed in ice-cold Hi-C lysis buffer (10 mM Tris-HCl pH 8.0, 10 mM NaCl, 0.2% Igepal CA630, 1x protease inhibitor cocktail) for 15 min. Pelleted nuclei were washed once with 1x NEBuffer 2 and incubated in 0.5% sodium dodecyl sulfate (SDS) at 62 °C for 5 min. After incubating, water and Triton X-100 were added to quench the SDS. MboI restriction enzyme (NEB, R0147) was then added and chromatin was digested. Biotin-14-dATP was used to mark the DNA ends followed by proximity ligation in intact nuclei. After crosslink reversal, samples were sheared to a length of ∼ 300 bp, and then treated with the End Repair/dA-Tailing Module (NEB, E7442L) and Ligation Module (NEB, E7445L) following the manufacturer's instructions. Biotin-labeled fragments were pulled down using Dynabeads MyOne Streptavidin T1 beads (Life technologies, 65602). The Hi-C library was amplified for about 11 cycles of PCR with Q5 master mix (NEB, M0492L) following the manufacturer's instructions. DNA was then purified, quantified, and sequenced from both ends using an Illumina sequencing platform.

### ChIP-seq library preparation

ChIP-seq was conducted according to previously described protocols [41] with few modifications. The cells were cross-linked with a final concentration of 1% formaldehyde followed by quenching with glycine. Cells were lysed with lysis buffer (0.2% SDS; 10 mM Tris-HCl, pH 8.0; 10 mM EDTA, pH 8.0; proteinase inhibitor cocktail) and sonicated to fragments about 300–500 bp (Bioruptor, Diagenode). Dynabeads Protein A was washed twice with ChIP Buffer (10 mM Tris-HCl pH 7.5, 140 mM NaCl, 1 mM EDTA, 0.5 mM EGTA, 1% Triton X-100, 0.1% SDS, 0.1% Na-deoxycholate, Cocktail proteinase inhibitor) and was incubated with antibody (ab8580 for H3K4me3, ab4729 for H3K27ac) at 4 °C for 2–3 h. The fragmented chromatin was transferred to the bead-antibody complex tubes and rotated at 4 °C overnight. The beads were washed once with low salt buffer (10 mM Tris-HCl pH 7.5, 250 mM NaCl, 1 mM EDTA, 0.5 mM EGTA, 1% Triton X-100, 0.1% SDS, 0.1% Na-deoxycholate, Cocktail proteinase inhibitor) and twice with high salt buffer (10 mM Tris-HCl pH 7.5, 500 mM NaCl, 1 mM EDTA, 0.5 mM EGTA, 1% Triton X-100, 0.1% SDS, 0.1% Na-deoxycholate, Cocktail proteinase inhibitor). After crosslink reversal, the library was constructed as Hi-C.

### Hi-C data processing

The quality of all libraries was assessed using FastQC. Reads with mean quality scores less than or equal to 30 were removed. Adapters were removed by cutadapt. Pig Hi-C reads were processed using a Juicer pipeline [42]. The data was aligned against the Sscrofa11.1 reference genome. Contact reads related with ChrY and ChrMT or with MAPQ = 0 were filtered out. The correlation between p(s) was assessed with Jensen-Shannon divergence (JSD). For each chromosome, we obtained the expected Hi-C contact values by calculating the average contact intensity for all loci at a certain distance. We then transformed the raw Hi-C matrix into an observed/expected (O/E) matrix by dividing each normalized observed value by its corresponding expected value. All

contact matrices used for further analysis were KR-normalized with Juicer. The correlation matrixes were calculated by Juicer. To compare the contact frequencies between different samples, we randomly sampled equal numbers of valid pairs from each sample.

### Analysis of A/B compartment

We used HOMER software to obtain the PC1 value and A/B compartment status [43]. We used 50 kb and 300 kb resolution to obtain the PC1 value in this assay. When comparing compartments with histone markers or genes, we defined the bin size as 50 kb. Except for PEFs, the A/B compartments were determined using the gene density of each chromosome. In PEFs, the A/B compartments were determined using the H3K27ac ChIP-Seq signal density. The compartment strength was calculated using $AA*BB/AB^2$ as previously described [44] with minor modification. To eliminate interference of interactions within TADs, we first removed local interactions shorter than 2 MB. For boxplots, we used the top 20% eigenvectors for A and the bottom 20% for B. Saddle plots were calculated as previously described with minor modifications [45]. Briefly, we first calculated O/E Hi-C matrices for 300 kb of binned data. Then, we sorted O/E Hi-C matrix bins according to the specific PC1 values and divided the matrices into 50 bins. Strength of compartmentalization was defined as AA (lower right corner)/AB (upper right corner) and BB (upper left corner)/BA (lower left corner). The values used for each corner were calculated as a mean value of 10 bins.

### A/B compartment conservation across species

A/B compartment status was identified at 50-kb resolution in PEFs and MEFs. Coordinates of 50-kb bins in PEFs were converted to mm10 using the UCSC Liftover tool with default parameters [46]. Bins were considered homologous if they overlapped more than 60% with each other. Then, we calculated proportion of homologous bins classified by A/B compartment status between PEFs and MEFs.

### Analysis of TADs

TADs were called using deDoc at best resolution for each chromosome [25]. ISs were calculated with bin size 50 kb [47]. To compare the IS between different samples, we randomly sampled equal numbers of valid pairs from each sample and calculated the IS. TAD similarity was calculated as previously described [25]. To compare TAD similarity between different stages, we used TADs called by deDoc with bin size 100 kb. For CTCF-motifs analysis, we counted the number of CTCF-motifs in forward or reverse orientation in ten 10 kb bins from the TAD-boundaries. TAD strength was calculated as previously described [44]. The heatmaps of insulation score for all samples were ordered by the IS of corresponding borders in PEF. And we plotted the IS profile in a 1.25 Mb region flanking the TAD borders at 50 kb resolution.

### TAD boundaries conservation across species

For TAD comparison between pig and mouse, pig TAD boundary coordinates were converted to mm10 using the UCSC Liftover tool with default parameters [46]. Note that only domains located in syntenic regions were successfully converted; thus, only

these domains were analyzed. After Liftover, TAD boundaries longer than 50 kb were discarded. Coordinates of TAD boundaries were considered similar if they overlapped with each other at least 1 bp. Randomly sampled genome loci were taken as controls.

To define conservation, we only considered the bins that could be successfully "liftover" to the mouse genome. For any given TAD in pigs, its conservation was defined as the percentage of intra-TAD bin pairs that could be liftover and the liftovered two bins also belonged to a TAD in mouse. Shuffled TADs were taken as controls for each chromosome. A bin was considered to belong to a TAD if 60% of its length overlapped with that TAD. To test the robustness of this definition, we adjusted the definition with three overlapping ratio, e.g., 0.7, 0.8, and 0.9, and found that the conservation level was seldom effected (Fig. S1F).

### Convergent CTCF loop identification

We used Fithic software to identified loops in PEFs and MEFs at 10 kb resolution [48]. We filtered results using a $q$ value $< 0.01$ for candidate loops. The resulted loops were then filtered based on CTCF ChIP-seq peaks and CTCF direction identified by MEME [49]. We discarded loops if the distance of loop anchors were less than 300 kb to avoid distance decay effects. Average APA scores were calculated by Juicer with a bin size of 10 kb.

To identify homologous CTCF loops in mouse, we converted the coordinates of both loop anchors in Sscrofa11.1 to mm10 by Liftover. Loop anchors longer than 15 kb were discarded. Loops were considered conserved if both converted anchors overlapped at least 1 bp with the de novo identified loop anchors in mouse.

### RNA-seq data analysis

RNA-seq data of PEFs were downloaded from GEO (SRR066366) and mapped to the pig reference genome (Sscrofa11.1) using HISAT2 with default parameters [50]. TPM values were obtained using StringTie [51].

### ChIP-Seq data analysis

All ChIP-Seq reads were mapped using bowtie2 [52]. Fragments with both ends uniquely mapped with MAPQ and larger than 5 are then extracted using samtools. Duplicates were removed by Picard tools MarkDuplicates. Peaks were called using the MACS2 callpeak command with a $q$ value of 0.01 [53]. Narrow peaks were called for all libraries. Fold enrichment over control signal tracks was built using the command bdgcmp in MACS2. Peaks called from all replicates of each condition were merged according to an irreproducible discovery rate (IDR) threshold of 0.01. Reads were then merged and a final fold enrichment over control track was made for each condition.

### Mouse data analysis

Mouse embryo Hi-C data and MEF Hi-C data were downloaded from GEO (GSE82185 and GSE121087, respectively). All mouse Hi-C reads were mapped to the mouse genome (mm10) using HiCUP pipeline v.0.5.7 [54]. We used SNPsplit software to distinguish paternal and maternal reads [55]. TADs were called by deDoc using a similar strategy as the pig data.

## Supplementary information

---

**Additional file 1.** Supplementary Text.

**Additional file 2: Fig. S1.** Reproducibility of biological replicates and conservation of chromatin architecture between PEF and MEF. **Fig. S2.** The prevalence of superdomains in pig preimplantation embryos. **Fig. S3.** Compartmentalization saddle plots of pig IVF and mouse Hi-C data. **Fig. S4.** Gradual establishment of TADs during pig IVF embryogenesis. **Fig. S5.** Loop reprogramming in pig and mouse preimplantation embryos. **Fig. S6.** Normalized Hi-C contact heatmaps for the same 10 Mb region as Fig. 3a for both replicates. **Fig. S7.** PC1 values and correlation matrix of chromosome 14 for both replicates and compartmentalization saddle plots of pig PA and AG Hi-C data. **Fig. S8.** Pearson correlation coefficient of the PC1 using single replicates. **Fig. S9.** PC1 and correlation matrices of all chromosomes for pig PA and AG embryos at different stages. **Fig. S10.** TADs are asynchronously established in the two parental alleles. **Fig. S11.** Embryonic development-related genes enriched in TAD boundaries with different insulation strengths compared uniparental with IVF embryos.

**Additional file 3: Table S1.** Summary of Hi-C and ChIP-seq data.

**Additional file 4: Table S2.** Comparison of the p(s) curve between uniparental embryos and the IVF embryos.

**Additional file 5.** Review history.

---

### Acknowledgements
Dr. Tracey Baas and Mr. David Martin performed English language editorial services.

### Review history
The review history is available as Additional file 5.

### Peer review information

### Authors' contributions
YW, ZZ, and JZ conceived this project. FL, XLL, and YJ performed the experiments, DW, BX, and XL analyzed data, RS, CC, YW, JH, QL, and NH collected the PEF and embryos, and YW, ZZ, JZ, FL, DW, CC, and XG prepared the manuscript. All authors read and approved the final manuscript.

### Funding
This work was supported by the Strategic Priority Research Program of the Chinese Academy of Sciences, China (XDA24020307, XDA16030101), the National Key R&D Program of China (2018YFC2000400), the National Transgenic Project of China (2016ZX08009003-006-007), the National Nature Science Foundation of China (31671342, 31871331, 91940304, 31672387, 81671274), and the Agricultural Science and Technology Innovation program of CAAS (ASTIP-IAS05).

### Availability of data and materials
The datasets generated during the current study are available in the Genome Sequence Archive in BIG Data Center (Big Data Center Members, 2017), Beijing Institute of Genomics (BIG), and Chinese Academy of Sciences, with accession numbers CRA002145 and CRA002146 (http://bigd.big.ac.cn/gsa/s/feQ4I5ET and http://bigd.big.ac.cn/gsa/s/JxP4c43Y), and also available at NCBI's Gene Expression Omnibus (GEO) with additional processed data files under accession number GSE153452 (https://www.ncbi.nlm.nih.gov/geo/query/acc.cgi?acc=GSE153452) [56].
Hi-C data of mouse preimplantation embryos and mouse embryonic fibroblasts (MEFs) were downloaded from the GEO database under the accession numbers of GSE82185 [16] and GSE121087 [23], respectively. RNA-seq data of PEFs were downloaded from the GEO under the accession number of SRR066366 [24].

### Ethics approval and consent to participate
Animal experiments were performed according to the guidelines of the Experimental Animal Welfare and Ethical of the Institute of Animal Science, Chinese Academy of Agricultural Sciences (IAS2018-11).

### Consent for publication
Not applicable.

### Competing interests
The authors declare that they have no competing interests.

### Author details
[1]CAS Key Laboratory of Genome Sciences and Information, Beijing Institute of Genomics, Chinese Academy of Sciences, and China National Center for Bioinformation, Beijing 100101, China. [2]University of Chinese Academy of Sciences, Beijing 100049, China. [3]State Key Laboratory of Stem Cell and Reproductive Biology, Institute of Zoology, Chinese Academy of Sciences, Beijing, China. [4]Guangdong Provincial Key Laboratory of Malignant Tumor Epigenetics and Gene Regulation, Medical Research Center, Sun Yat-Sen Memorial Hospital, Sun Yat-Sen University, Guangzhou 510120, China. [5]Institute of Animal Science, Chinese Academy of Agricultural Sciences, Beijing 100193, China.

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

## 
