## [**Additional file 5.** Review history. · Genome Biology]

Review History

First round of review

Reviewer 1

Are you able to assess all statistics in the manuscript, including the appropriateness of statistical tests used? Yes, and I have assessed the statistics in my report.

Comments to author:

Li et al present a manuscript in which they describe Hi-C data in pig embryonic fibroblasts and in early pig development. As expected they find the same features of 3D genome organization in pigs as in other vertebrates/mammals. During development they show very similar phenotypes as in mouse and human, i.e. no TADs in early development with TADs becoming gradually more prominent at later stages. Next they go on to show how the 3D genome develops during early embryogenesis in uniparental embryos. Here they find that during the 4 cell stage compartmentalization becomes weaker, something not seen in IVF embryos. Beyond this I had difficulty to identify novel findings in part because of the poor representation of the data and low resolution of some of the figures.

The authors show mostly abstract statistics of their Hi-C data. They should use more examples to strengthen their observations. For instance Figure 4A is interesting, but this is shown only for one chromosome. I would like to see this for more chromosomes (show a Supp Figure with all chromosomes).

Furthermore the representation of the statistics is often very crowded which makes it nearly impossible to understand what is being presented. The authors should make an effort to make the data intelligible for the general reader. The authors should make saddle plots to represent the compartment data.

Finally the authors claim that the difference in 3D genome between uniparental and IVF embryos may lead to a higher embryonic lethality. This is highly speculative and the analysis of imprinted genes is correlative at best. The authors should remove these claims or substantiate them with functional data.

Other points:

* With regard to reproducibility, the authors should use a method such as GenomeDISCO to properly calculate the reproducibility of the samples. They should also show for some of their key figures (i.e. 3A, 4A) what the replicates look like.

* p.4 l. 5 cohesion should be cohesin

* p. 4 l. 13 in medaka there is no structure during ZGA. Zebrafish and medaka differ in that respect from mammals and drosophila.

* For the average insulation score plots the authors should also show heatmaps.

* p. 5 l. 17 ($p < 0.045$) I assume $p = 0.045$ is meant

* At what resolution was the TAD conservation analysis performed. It would be good to know how the authors treat off-by-one results.

- * p. 6 l. 2 "homologous bin pair": how is this analysis performed, what is measured?
- * The compartment strength is defined as $(AA*BB)/AB^2$, the authors should recalculate the compartment strengths and remake the figures.
- * p. 8, l. 4: "Furthermore, the overall arrangement of TADs gradually approaches the patterns seen in PEFs." What is this statement based on?
- * p. 9 "In addition to the local contact, the progress of distal ($> 10 M$) and intermediate contacts ($< 1 M$ and $> 50 K$) are also significant different in the uniparental embryos from the IVF embryos." What is the statistical test to claim significance.
- * Figure 4A show scatter plots comparing the compartment scores.
- * Figure 4C show the correlations between replicates + scatterplots
- * Figure 6A+B are very unclear
- * Supposedly the text has been read by a native speaker, but there are still some obvious mistakes, i.e. imprinting genes where imprinted genes is meant or fatal embryo where embryonic lethality is meant.

Reviewer 2

Are you able to assess all statistics in the manuscript, including the appropriateness of statistical tests used? Yes, and I have assessed the statistics in my report.

Comments to author:

The asynchronous establishment of chromatin 3D architecture between in vitro fertilized and uniparental preimplantation pig embryos by Li et al.

This is an interesting manuscript that examines the establishment of chromatin conformation during early embryonic in pig. The examination of this experimental system is interesting since it seems that subtle differences in how chromatin conformation appears at the onset of zygotic genome activation between vertebrates. Therefore, the biological question is of significant importance. To tackle this question, the authors first obtain chromatin conformation maps of pig embryonic fibroblasts and compare them with those from mouse embryonic fibroblasts (Fig 1), demonstrating that chromatin conformation in pigs have a similar organisation into compartments, topologically associating domains (TADs) and loops, and that some of the loops might even be conserved between the two species. The authors then examine the emergence of chromatin conformation during early embryonic development by generating chromatin conformation maps of zygotes, 4-cell embryos and morula embryos produced using IVF techniques. Similarly as it has been observed before in mouse and flies, the authors report a very weak level of chromatin conformation for zygote and 4-cell stage and a progressive acquisition of chromatin conformation from morula onwards (Fig 2). The authors then compare the chromatin conformation dynamics of IVF embryos with those obtained from parthenogenesis (PA) and androgenesis (AG) (Fig 3, 4 and 5). The authors suggest that there are different chromatin conformation dynamics for these embryos compared with IVF embryos. The authors then evaluate maternal/paternal differences in mouse embryos and suggest that there are different dynamics in how these are established

(Fig 5D and 5E) and then link the observation with an evaluation of the changes of insulation score at regions that change chromatin conformation in imprinted genes (Fig 6).

Overall the manuscript is potentially of high interest. However, I have serious concerns about the quality of some of the Hi-C datasets compared here, which casts serious doubts regarding the level of support for the conclusions reached by the authors that I will review in turn.

Major points:

1. Quality of 4-cell PA embryos and 4-cell and morula AG embryos

The authors report significant differences in chromatin organisation for 4-cell PA embryos and 4-cell and morula AG embryos. However, the sequencing statistics provided by the authors show that those datasets are technically very different from the other datasets generated in this study. A preliminary analysis of some of the data provided by the authors shows, in agreement with the statistics included in Supp. Table 1, that a significant fraction of those datasets are PCR duplicates. For example, in the 4-cell AG embryos, from an initial amount of ~420 million reads, less than 5% of those (~25 million) are used for the analysis. The differences in sequence coverage appear for all three datasets for which the authors detect differences. Therefore, without additional analysis, and since sequencing depth is a major determinant for the characterisation of compartments and calculation of insulation score, it is not possible to determine whether the reported differences are just due to the technical differences (rather than the biological differences) between these datasets. This severely affects the results presented in Fig 3, 4, 5 and 6, since these are based on the calculation of magnitudes (insulation scores and correlation coefficients) from datasets with very sparse information and therefore, it is not possible to determine whether the reported changes are relevant biologically, or technical artefacts.

2. In order to examine the effect of the sequencing depth in these datasets, the authors could evaluate whether there are any differences between the two replicates of PA morula, since both have the same level of initial sequencing depth, but the amount of valid pairs resulting from these samples are very different (50% for replicated one vs 5% for replicate two). Contact frequency distribution, insulation score analysis and eigenvector decomposition of the cross-correlation matrices between these two biological similar samples will reveal the effect of differential sequencing depth in these analyses. Alternatively, the authors could downsample samples with higher level of sequencing depth to control for the effect in the downstream analyses.

3. A similar analysis should be performed for the maternal and paternal mouse Hi-C datasets, since significant differences in detecting maternal vs paternal alleles will have a confounding effect in the calculation of allele specific insulation scores.

4. The effect of data sparsity should also be examined within the context of the contact frequency analysis presented in Figs 2, 3 and 5, since KR-balancing of sparse matrices can have a severe effect on the distribution of contacts.

5. The authors should also include 95-percentile confident intervals to the mean insulation plots presented in the study. This would be particularly useful in Fig 5B, since the reported changes in mean insulation scores are very weak in magnitude compare for example with the level of insulation at TAD borders in PEFs (Fig 2G).

6. I do not understand the reasoning behind the authors conclusion that compartmentalisation is established differently in pig and mouse embryos (line 5-25; page 7). In my opinion, the PC1 eigenvector in Fig 2C shows a clear separation in chromosomal arms for zygote, 4-cell and morula pig embryos, suggesting that a finer compartmentalisation of the genome is not present at these stages. The level of compartmentalisation in mouse embryos at that stage is also very limited. The authors should generate saddle plots both for the pig and mouse samples to examine these differences systematically.

7. The changes in insulation score for imprinted genes in Fig 6 are difficult to interpret based on the current limitations of the dataset (see point 1). Once these limitations have been addressed, it would be useful if the authors could show zoom in plots for the Hi-C maps to display the observed changes in chromatin conformation at these stages and whether TAD borders change at these locations during development.

GBIO-D-20-00090

The asynchronous establishment of chromatin 3D architecture between in vitro fertilized and uniparental preimplantation pig embryos

Feifei Li; Danyang Wang; Ruigao Song; Chunwei Cao; Zhihua Zhang, Ph.D; Yu Wang; Xiaoli Li; Jiaojiao Huang; Qiang Liu; Naipeng Hou; Bingxiang Xu; Xiao Li; Xiaomeng Gao; Yan Jia; Jianguo Zhao; Yanfang Wang Genome Biology

Reviewer #1: Li et al present a manuscript in which they describe Hi-C data in pig embryonic fibroblasts and in early pig development. As expected they find the same features of 3D genome organization in pigs as in other vertebrates/mammals. During development they show very similar phenotypes as in mouse and human, i.e. no TADs in early development with TADs becoming gradually more prominent at later stages. Next they go on to show how the 3D genome develops during early embryogenesis in uniparental embryos. Here they find that during the 4 cell stage compartmentalization becomes weaker, something not seen in IVF embryos. Beyond this I had difficulty to identify novel findings in part because of the poor representation of the data and low resolution of some of the figures.

1. The authors show mostly abstract statistics of their Hi-C data. They should use more examples to strengthen their observations. For instance Figure 4A is interesting, but this is shown only for one chromosome. I would like to see this for more chromosomes (show a Supp Figure with all chromosomes).

Answer: Thank you for the suggestion. We have added saddle plots to Figure 4A to quantitatively demonstrate the decompartmentalization in the 4-cell stage of both PA and AG. The correlation matrices for all chromosomes can be found in Fig S7, and we also pasted the matrix heatmaps below. We can see that the correlation matrices showed decompartmentalization in both PA and AG embryos at the 4-cell stage for most chromosomes, except for chr13 and chr15. The decompartmentalization can also be seen quantitatively from the saddle plots (Fig S6) and compartment strength (Fig 4B in main text). As shown in Fig4B, the compartment strength of AG at the 4-cell stage is significantly lower than that in the zygote and morula stage. It's also significantly lower in PA 4-cell stage than that in the PA morula, and moderate lower than the PA zygote ($p=0.065$, Mann-Whitney U-test, one-tailed test). Considering the total chromosome number is 19 in pig, the $P=0.065$ means that the chance of making a false statement of "*the compartment strength in PA 4cell is lower than that in zygote*" is about in only one chromosome. Thus, we believe the conclusion is solid.

Fig S7. PC1 and correlation matrices of all chromosomes for pig PA and AG embryos.

Fig S6. Compartmentalization saddle plots of pig Hi-C data for PA and AG embryos (binsize = 300kb).

2. Furthermore the representation of the statistics is often very crowded which makes it nearly impossible to understand what is being presented. The authors should make an effort to make the data intelligible for the general reader. The authors should make saddle plots to represent the compartment data.

Answer: Thank you for the suggestion. In line with this comment, several figures were re-organized. We changed Fig 1J to a cumulative curve to clearly show the difference between the experimental data and the control, and moved Fig 5E to the supplementary figures as it is less informative. We also reorganized Fig6A and Fig6B

into 12 subpanels with each subpanel showing only one type of pattern, e.g., the promoter region of MKRN3 and TRAPPC9 have higher IS in PA embryos than IVF embryos in all three developmental stages. Finally, for all the compartment data, we have added saddle plots, including FigS3 and S6. The numbers in the corners represent the strength of the AA interactions as compared to the AB interactions and the BB interactions over the BA interactions. Overall, the results of the saddle plots support our conclusions of the gradually established compartmentalization during IVF embryo development, especially after ZGA. The decompartmentalization is also shown in both PA and AG embryos at the 4-cell stage (Fig S6). We also include saddle plots for mouse embryos, and the results show that the compartmentalization of mouse embryos is increasingly strong (Fig S3B).

Fig S3. Compartmentalization saddle plots of pig IVF (A) and mouse (B) Hi-C data binned at 300kb resolution at different developmental stages.

Fig S6. Compartmentalization saddle plots of pig PA and AG Hi-C data binned at 300kb resolution at different developmental stages.

3. Finally the authors claim that the difference in 3D genome between uniparental and IVF embryos may lead to a higher embryonic lethality. This is highly speculative and the analysis of imprinted genes is correlative at best. The authors should remove these claims or substantiate them with functional data.

Answer: Thanks for the comment. We acknowledge that the link between the unusual chromatin pattern we observed in uniparental embryos and the higher embryonic lethality is not directional, and a functional assay will be needed to substantiate them. However, the in vitro production (IVP) of porcine embryos has presented numerous challenges to researchers over the past few decades. Compared with other model or livestock animals, such as cattle and sheep, pig IVP has not been fully developed due to several technical problems, such as incomplete final maturation status after in vitro maturation, a high incidence of polyspermy after in vitro fertilization, and low development rate and poor quality of blastocysts after culturing (PMID: 12467351 and 24274407). These limitations in porcine IVP add numerous challenges for sample collection. Thus, given that the pig is a much less developed model and has a longer gestation period, and most of the genetic tools are not available for this system, the task to perform such an assay is currently far beyond our capacity. Thus, we toned down the claim in the main text accordingly.

Other points:

4. * With regard to reproducibility, the authors should use a method such as GenomeDISCO to properly calculate the reproducibility of the samples.

Answer: Thank you for the suggestion. According to a published comparison of algorithms on Hi-C reproducibility assessment (PMID: 30890172), GenomeDISCO is sensitive to distal contact and the assessment can be tricky, while HiCRep is less sensitive to distal contact and has been widely used for assessment. Therefore, in addition to GenomeDISCO, we also calculated the reproducibility of samples using HiCRep. The contact matrix we produced was relatively sparse, which is common for libraries with rare samples, such as early-development embryos. HiCRep explicitly corrects for the genomic distance effect and addresses the sparsity of contact matrices through stratification and smoothing (PMID: 30890172), which is more suitable for our data. Nevertheless, both HiCRep and GenomeDISCO reported high reproducibility in most libraries (see Figures below). Although the GenomeDISCO correlation in the IVF morula was found to be less significant, the following evidences supported that the reproducibility of IVF morula is sufficiently robust for the analysis we present in the manuscript. First, as determined by HiCRep, the reproducibility of the IVF morula is perfect, and the mean SCC of the IVF morula is 0.914, which is the fourth highest score among all ten samples. Second, as mentioned above, the GenomeDISCO is sensitive to the distal genomic contacts (PMID: 29554289). However, the features we discussed in the manuscript were almost relative local, i.e. $< 10M$. For example, all TADs are smaller than 10M, and distal contacts contribute little to the PC1 in the compartment analysis due to the sparse

nature of the Hi-C data. Moreover, we found that the GenomeDISCO correlation between the replicates in the IVF morula library is significantly greater than that between the any two randomly picked samples (FigS1B below). Together, the reproducibility of our libraries are sufficiently greater than what the current analysis demanded.

Fig S1A and S1B. Reproducibility of biological replicates calculated using HiCRep and GenomeDISCO. Correlation between the examined sample and one randomly picked sample was used as control and shown in grey in GenomeDISCO.

5. They should also show for some of their key figures (i.e. 3A, 4A) what the replicates look like.

Answer: Thanks for the suggestion. We show the results of two replicates for the chromosome in Fig3A and 4A, and Fig I and Fig II below. It is clear that the two replicates are rather similar to each other. We further demonstrate the reproducibility by aggregate analysis. As suggested by reviewer 2, we used downsampled data in this revision, which resulted in a relatively sparse Hi-C contact map and correlation matrix. However, even so, the aggregate TAD signal and saddle plots of the replicates showed high similarity to each other.

Fig I. Normalized Hi-C contact heatmaps for the same 10 Mb region as figure3A and the aggregate TAD signal were shown for both replicates at different developmental stages for PA and AG embryos.

Fig II. PC1 values and correlation matrix for the same chromosome of figure 4A and the compartment saddle plots were shown for both replicates at different developmental stages.

6. p.4 l. 5 cohesion should be cohesin
Answer: Thanks, we corrected this typo.

7. * p. 4 l. 13 in medaka there is no structure during ZGA. Zebrafish and medaka differ in that respect from mammals and drosophila.

Answer: Thanks. In a bioRxiv paper ‘CTCF looping is established during gastrulation in medaka embryos’ (DOI: 10.1101/454082), the authors stated that ‘Starting at 7 hpf - the middle of zygotic genome activation– we observed the emergence of a plaid pattern, consistent with the presence of two compartments’. The authors also stated that contact domains also began to be visible during ZGA, although the domain size is small and becomes much larger during gastrulation. Since the bioRxiv paper has not been peer reviewed, the conclusion may be subject to change. Therefore, we modified our statement, and clearly state that the pattern was reported in a preprint paper.

8. * For the average insulation score plots the authors should also show heatmaps.

Answer: Thank you for the suggestion. We added the heatmaps of all TAD boundaries into FigS4A and S8A. The heatmaps support our conclusions that the TAD structures are gradually established in all three kinds of embryos but are asynchronously established in PA and AG embryos.

Fig S4A. Heatmaps showing the strength of all TAD boundaries at different developmental stages of pig IVF embryos.

A

Fig S8A. Heatmaps showing the strength of all TAD boundaries at different developmental stages of pig PA and AG embryos.

9. * p. 5 l. 17 ($p < 0.045$) I assume $p = 0.045$ is meant

Answer: Thanks, we corrected it.

10. * At what resolution was the TAD conservation analysis performed. It would be good to know how the authors treat off-by-one results.

Answer: Thanks for the question. The resolution for the analysis of TAD conservation was 40kb. To define conservation, we only considered the bins that could be successfully “liftover” to the mouse genome (UCSC Liftover, see the next answer). Let’s denote those bins as a set B. For any given TAD in pigs, its conservation was defined as the percentage of intra-TAD bin pairs in B that could be liftover and the liftovered two bins also belonged to a TAD in mouse. A bin was considered belong to a TAD if 60% of its length overlapped with that TAD. To test the robustness of this definition, we adjusted the definition with three overlapping ratio, e.g., 0.7, 0.8 and 0.9, and found that the conservation level was seldom effected (Fig III). Thus, off-by-one error should not be a serious concern.

Fig III. Cumulative curve of TAD according to TAD conservation. The same ratio under randomly shuffled TADs was shown in gray as control. Four panels were drawn for different overlapping ratio of a bin belonging to one TAD.

11. * p. 6 l. 2 "homologous bin pair": how is this analysis performed, what is measured?

Answer: Thanks. The homologous bin in mouse was defined using the UCSC Liftover tool, i.e., if both two bins in pig can be successfully liftovered into two independent bins in mouse.

12. * The compartment strength is defined as $(AA*BB)/AB^2$, the authors should recalculate the compartment strengths and remake the figures.

Answer: Thanks for the suggestion. We recalculated the compartment strength using the formula that the reviewer provided at 300kb resolution and remade the figures. The new results support our conclusions of gradually enhanced segregation level during IVF embryo development, especially after ZGA, and the decompartmentalization in both PA and AG embryos at the 4-cell stage. Considering the total chromosome number is 19 in pigs, the P value between 0.05-0.1 means that the chance of making a false conclusion would only occur in about one or two chromosomes. We labeled the p value between 0.05-0.1 as triangles in the figure.

Fig 2G and 4B. Compartment strength after changing the calculation formula.

13. * p. 8, l. 4: "Furthermore, the overall arrangement of TADs gradually approaches the patterns seen in PEFs." What is this statement based on?

Answer: Thanks for the question. We measured the similarity of two TAD sets as previously described (PMID: 30111883). By comparing the TADs in PEFs to the TADs in morula, 4-cell stage and zygote, we found that the similarities gradually decrease (Fig. S4B), i.e., the further the development stage is from PEFs, the less TAD similarity one could expect between the two samples. Furthermore, TAD structure were previous considered conserved between developed tissue/cell-types, and we can take the TAD structure in PEFs as the generic goal for TAD development. Thus, we believe this gradually decreased TAD similarities implying a gradually approaches towards the formation of TAD in PEF.

14. * p. 9 "In addition to the local contact, the progress of distal (> 10 M) and intermediate contacts (< 1 M and > 50 K) are also significant different in the uniparental embryos from the IVF embryos." What is the statistical test to claim significance.

Answer: Thanks for the question. The statistical test we used was a t-test for the JSD in the range of the p(s) curve between uniparental embryos and the IVF embryos. The background of JSD distribution were estimated using the JSDs between replicates. By doing this test, we found that more than half the samples were significantly different ($P < 0.05$, Table S2). As not all samples showed the feature, we toned down our statement and used the sentence "In addition to the local contact, the progress of distal (> 10 M) and intermediate contacts (< 1 M and > 50 K) are also significantly different in more than half of the samples when comparing uniparental embryos to IVF embryos."

Table S2. Comparison of the p(s) curve between uniparental embryos and the IVF embryos

Large than 10 MB				
samples	JSD of Uniparental and IVF	Mean JSD of replicates	P value	P value <0.05
PA_zyg	0.0029	0.0005	1.5e-06	√
PA_4cell	0.0003	0.0005	0.377	
PA_morula	0.0004	0.0005	0.700	
AG_zyg	0.0008	0.0005	0.155	
AG_4cell	0.0010	0.0005	0.030	√
AG_morula	0.0013	0.0005	0.004	√

50KB ~ 1 MB				
samples	JSD of Uniparental and IVF	Mean JSD of replicates	P value	P value <0.05
PA_zyg	0.0056	0.0006	0.0	√
PA_4cell	0.0024	0.0006	2.94e-05	√
PA_morula	0.0023	0.0006	6.84e-05	√
AG_zyg	0.0005	0.0006	0.491	
AG_4cell	0.0018	0.0006	0.0008	√
AG_morula	0.0001	0.0006	0.074	

15. * Figure 4A show scatter plots comparing the compartment scores.

Answer: Thank you for the suggestion. We show the scatter plots below, which show that for nearly all chromosomes, the compartment scores of the 4-cell stage are lower than that of the zygote and morula for both PA and AG embryos, indicating decompartmentalization during the 4-cell stage (Fig IV).

Fig IV. Scatter plots showing the compartment scores of all chromosomes in PA and AG preimplantation embryos.

16. * Figure 4C show the correlations between replicates + scatterplots

Answer: Thanks for the suggestion. The figure below shows the correlation of PC1 for both replicates. The results showed that the drastic decrease in the PCC of the PC1 for both IVF-vs-uniparental embryos and maternal-vs-paternal embryos comparisons at the 4-cell stages exist in both replicates (Fig V). We also showed the scatterplots of the PCC using replicates, which highlighted a consistent lower PCC for all the comparisons in the 4-cell stage than the zygotes and morula (Fig VI).

Fig V. Boxplots show Pearson correlation coefficient of the PC1 between replicates.

Fig VI. Scatter plots of PCC between samples using single replicates.

17. * Figure 6A+B are very unclear

Answer: Thanks. We rearranged the plots by showing the difference of IS scores between uniparental and IVF embryos for the identified imprinted genes. And we classified these genes according to the changing pattern during development (Figure 6A and 6B).

Fig 6A and B. Differences of insulation scores between uniparental and IVF embryos for the identified differential imprinted genes. Genes are classified according to the changing pattern during development.

18. * Supposedly the text has been read by a native speaker, but there are still some obvious mistakes, i.e. imprinting genes where imprinted genes is meant or fatal embryo where embryonic lethality is meant.

Answer: Thank you for pointing these. We have asked for additional language proof reading from a native speaker.

Reviewer #2: The asynchronous establishment of chromatin 3D architecture between in vitro fertilized and uniparental preimplantation pig embryos by Li et al.

This is an interesting manuscript that examines the establishment of chromatin conformation during early embryonic in pig. The examination of this experimental system is interesting since it seems that subtle differences in how chromatin conformation appears at the onset of zygotic genome activation between vertebrates. Therefore, the biological question is of significant importance. To tackle this question, the authors first obtain chromatin conformation maps of pig embryonic fibroblasts and compare them with those from mouse embryonic fibroblasts (Fig 1), demonstrating that chromatin conformation in pigs have a similar organization into compartments, topologically associating domains (TADs) and loops, and that some of the loops might even be conserved between the two species. The authors then examine the emergence of chromatin conformation during early embryonic development by generating chromatin conformation maps of zygotes, 4-cell embryos and morula embryos produced using IVF techniques. Similarly, as it has been observed before in mouse and flies, the authors report a very weak level of chromatin conformation for zygote and 4-cell stage and a progressive acquisition of chromatin conformation from morula onwards (Fig 2). The authors then compare the chromatin conformation dynamics of IVF embryos with those obtained from parthenogenesis (PA) and androgenesis (AG) (Fig 3, 4 and 5). The authors suggest that there are different chromatin conformation dynamics for these embryos compared with IVF embryos. The authors then evaluate maternal/paternal differences in mouse embryos and suggest that there are different dynamics in how these are established (Fig 5D and 5E) and then link the observation with an evaluation of the changes of insulation score at regions that change chromatin conformation in imprinted genes (Fig 6).

Overall the manuscript is potentially of high interest. However, I have serious concerns about the quality of some of the Hi-C datasets compared here, which casts serious doubts regarding the level of support for the conclusions reached by the authors that I will review in turn.

Major points:

1. Quality of 4-cell PA embryos and 4-cell and morula AG embryos The authors report significant differences in chromatin organisation for 4-cell PA embryos and 4-cell and morula AG embryos. However, the sequencing statistics provided by the authors show that those datasets are technically very different from the other datasets generated in this study. A preliminary analysis of some of the data provided by the authors shows, in agreement with the statistics included in Supp. Table 1, that a significant fraction of those datasets are PCR duplicates. For example, in the 4-cell AG embryos, from an initial amount of ~420 million reads, less than 5% of those (~25 million) are used for the analysis. The differences in sequence coverage appear for all three datasets for which the authors detect

differences. Therefore, without additional analysis, and since sequencing depth is a major determinant for the characterisation of compartments and calculation of insulation score, it is not possible to determine whether the reported differences are just due to the technical differences (rather than the biological differences) between these datasets. This severely affects the results presented in Fig 3, 4, 5 and 6, since these are based on the calculation of magnitudes (insulation scores and correlation coefficients) from datasets with very sparse information and therefore, it is not possible to determine whether the reported changes are relevant biologically, or technical artefacts.

Answer: Thank you. We also noticed the differences in the valid read numbers between samples in the libraries. By conducting several examinations, we show that the data quality of those valid reads is sufficient and has little effect on the data analysis. We used the suggestion of our reviewers to guide our comparisons and analyses, and we appreciated their suggestions.

First, we analyzed the reproducibility of the libraries at the Hi-C contact map level using HiCRep and GenomeDISCO software. Both software packages reported high reproducibility for almost all the samples (Fig S1A and S1B). The contact matrix we produced was relatively sparse, which is common for libraries with rare samples, such as early-development embryos. HiCRep explicitly corrects for the genomic distance effect and addresses the sparsity of contact matrices through stratification and smoothing (PMID: 30890172), which is more suitable for our data. Nevertheless, both HiCRep and GenomeDISCO reported high reproducibility in most libraries (see Figures below). Although the GenomeDISCO correlation in the IVF morula was found to be less significant, the following evidences supported that the reproducibility of IVF morula is sufficiently robust for the analysis we present in the manuscript. First, as determined by HiCRep, the reproducibility of the IVF morula is perfect, and the mean SCC of the IVF morula is 0.914, which is the fourth highest score among all ten samples. Second, as mentioned above, the GenomeDISCO is sensitive to the distal genomic contacts (PMID: 29554289). However, the features we discussed in the manuscript were almost relative local, i.e. < 10M. For example, all TADs are smaller than 10M, and distal contacts contribute little to the PC1 in the compartment analysis due to the sparse nature of the Hi-C data. Moreover, we found that the GenomeDISCO correlation between the replicates in the IVF morula library is significantly greater than that between the any two randomly picked samples (FigS1B below). Together, the reproducibility of our libraries are sufficiently greater than what the current analysis demanded.

Fig S1A and S1B. Reproducibility of biological replicates calculated using HiCRep and GenomeDISCO. Correlation between the examined sample and one randomly picked sample was used as control and shown in grey in GenomeDISCO.

Second, we revised the manuscript with all the comparisons using downsampled data. In another words, p(s) curves, IS, compartments and correlations were all recalculated with the same level of data coverage, which was equivalent to the AG 4-cell stage. The revised analysis almost completely recaptured the patterns we reported in the previous analysis, albeit with some minor modifications. For example, the p(s) curves were almost identical to the previous curves for both pig and mouse data. This is also supported by our answers to your question No. 4 and shown in our updated Fig 2F, 3B and 3C. For the compartments, the compartment strength was recalculated using the formula that the reviewer1 provided, e.g. $(AA*BB)/AB^2$, at 300kb resolution. The new results support our conclusions of gradually enhanced segregation level during IVF embryo development, especially after ZGA, and the decompartmentalization in both PA and AG embryos at the 4-cell stage, as shown in our updated Fig 2G and 4B. Considering the total chromosome number is 19 in pigs, the P value between 0.05-0.1 means that the chance of making a false conclusion would only occur in about one or two chromosomes. We labeled the p value between 0.05-0.1 as triangles in the figure. For TAD, gradual establishment was also found in all IVF, PA and GA embryos, and asynchrony remained intact between PA and AG (Fig VII).

Fig 2G and 4B. Compartment strength after changing the calculation formula.

Fig VII. Comparisons of observed/expected (O/E) aggregate plot of TADs and mean value of IS around TAD borders before (left column) and after (right column) downsampling data for both IVF and uniparental pig embryos.

2. In order to examine the effect of the sequencing depth in these datasets, the authors could evaluate whether there are any differences between the two replicates of PA morula, since both have the same level of initial sequencing depth, but the amount of valid pairs resulting from these samples are very different (50% for replicated one vs 5% for replicate two). Contact frequency distribution, insulation score analysis and eigenvector decomposition of the cross-correlation matrices between these two biological similar samples will reveal the effect of differential sequencing depth in these analyses. Alternatively, the authors could downsample samples with higher level of sequencing depth to control for the effect in the downstream analyses.

Answer: Thank you for the suggestion. As mentioned in our answer to your first question, the reproducibility of the Hi-C contact map of valid reads was sufficient in the libraries. In this revised manuscript, we pooled data from replicates, and downsampled reads of all samples into one with the lowest coverage, following your suggestion. We showed that the pattern between those downsampled libraries were rather consistent (again, see our answer to your first question).

3. A similar analysis should be performed for the maternal and paternal mouse Hi-C datasets, since significant differences in detecting maternal vs paternal alleles will have a confounding effect in the calculation of allele specific insulation scores.

Answer: Thank you for the suggestion. We revised Fig 3C and 5D using the downsampling strategy. Our results showed that p(s) curves are almost identical to the previous curves. The IS for maternal alleles changed a little, but the pattern that TAD establishment is faster in maternal chromosomes than those in paternal chromosomes remains, except for the 8-cell stage. We highlight this difference in the main text.

Fig 3C and 5D. The contact frequency decay curves of mouse Hi-C data and mean values of insulation scores at MEF borders in maternal (left) and paternal (right) alleles were shown under the downsampled data.

4. The effect of data sparsity should also be examined within the context of the contact frequency analysis presented in Figs 2, 3 and 5, since KR-balancing of sparse matrices can have a severe effect on the distribution of contacts.

Answer: Thank you for the suggestion. We revised the p(s) curves using the downsampled data and the results showed that the pattern is identical to the previous version for both pig and mouse data (Figure below).

Fig 2F. The p(s) curves using the downsampled data for both pig IVF and mouse embryos.

Fig 3B and 3C. The $p(s)$ curves using the downsampled data for pig PA and AG embryos as well as mouse maternal and paternal alleles.

5. The authors should also include 95-percentile confident intervals to the mean insulation plots presented in the study. This would be particularly useful in Fig 5B, since the reported changes in mean insulation scores are very weak in magnitude compare for example with the level of insulation at TAD borders in PEFs (Fig 2G).

Answer: Thank you for the suggestion. The difference in the mean insulation score for uniparental embryos compared with IVF embryos is indeed minor. We think including 95-percentile confident intervals might make the plot too crowded to see the primary information. Thus, we alternatively drawn heatmaps of all TAD boundaries to show variations. From the heatmaps, we can also identify the pattern that the TAD structures asynchronously established in PA and AG embryos (Fig S4A and S8A).

Fig S4A. Heatmaps showing the strength of all TAD boundaries at different developmental stages of pig IVF embryos.

Fig S8A. Heatmaps showing the strength of all TAD boundaries at different developmental stages of pig PA and AG embryos.

6. I do not understand the reasoning behind the authors conclusion that compartmentalisation is established differently in pig and mouse embryos (line 5-25; page 7). In my opinion, the PC1 eigenvector in Fig 2C shows a clear separation in chromosomal arms for zygote, 4-cell and morula pig embryos, suggesting that a finer compartmentalisation of the genome is not present at these stages. The level of compartmentalisation in mouse embryos at that stage is also very limited. The authors should generate saddle plots both for the pig and mouse samples to examine these differences systematically.

Answer: Thank you for the suggestion. We now show the PC1 eigenvector and correlation matrix for all chromosomes (Fig S2A). We agreed with the reviewer that in some chromosomes the PC1 eigenvector showed a clear separation in chromosomal arms. In addition, by visual inspection of the PC1 of Hi-C matrix in pig preimplantation embryos (Fig S2A), we noticed that many chromosomes were made up of only a few superdomains, where the compartment domain was larger than 10Mb. For example, three superdomains almost covered all of chr6 (Fig S2A). For chr3, chr4, chr8 and chr10, the chromosome arms could be clearly separated by the PC1 (Fig S2A). However, such superdomains were much less prevalent in mouse preimplantation embryos. We quantified the prevalence of those superdomains using the accumulative curve of genome coverage as a function of domain size (Fig 2C).

For segregation level, although the segregation levels of chromosome in pig embryos, according to compartment strength and saddle plots, increased from zygotes to morula, the levels were very different from what was observed in PEFs (Fig 2D and S3). These data suggest that a finer compartmentalization of the genome is not present until the morula stage. This is similar to mouse in which the segregation level of 8-cell embryos is also lower than that in mouse ICM cells.

In addition, for all our compartment data, we have added saddle plots, including FigS3 and S6. The numbers in the corners represent the strength of AA interactions as compared to AB interaction and BB interactions as compared to BA interactions. Overall, the results of the saddle plots support our conclusions of the gradually established compartmentalization during IVF embryo development, especially after ZGA. The decompartmentalization is also shown in both PA and AG embryos at the 4-cell stage (Fig S6). We also include saddle plots for mouse embryos, and the results show that the compartmentalization of mouse embryos is increasingly strong (Fig S3B).

Thus, we rewrote the paragraph to make the following points. First, gradual compartmentalization occurs during IVF embryo development, especially after ZGA. However, compartmentalization have not be fully established. Second, pig preimplantation embryos contain a prevalence of superdomains.

A

Fig S2A. PC1 and correlation matrices of all chromosomes for IVF embryos.

Fig 2C. The accumulative curve for genome coverage by domains size. X-axis represents the relative domain size, ie. domain length /chromosome length. Only domains larger than 1MB were included in this figure. The P-values are given for the t-test of average domain sizes.

Fig S3. Compartmentalization saddle plots of pig IVF (A) and mouse (B) Hi-C data binned at 300kb resolution at different developmental stages.

Fig S6. Compartmentalization saddle plots of pig PA and AG Hi-C data binned at 300kb resolution at different developmental stages.

7. The changes in insulation score for imprinted genes in Fig 6 are difficult to interpret based on the current limitations of the dataset (see point 1). Once these limitations have been addressed, it would be useful if the authors could show zoom in plots for the Hi-C maps to display the observed changes in chromatin conformation at these stages and whether TAD borders change at these locations during development.

Answer: Thank you for the suggestion. We downsampled the data to control for the

effect of sequencing depth in our analysis. Using these data, we show the differences in the IS score between uniparental and IVF embryos for the identified imprinted genes. Next, we classified these genes according to the changing pattern during development (Fig 6A and 6B, shown below). Finally, we include a high-resolution contact heatmap of the *KCNQ1* gene (Fig VIII), which shows stronger insulation in the PA zygote and PA morula than the corresponding IVF embryos, and was recently reported to be important during mouse uniparental reproduction. Although the contact heatmap is relatively sparse, the trend can still be seen.

Fig 6A and B. Differences of insulation scores between uniparental and IVF embryos for the identified differential imprinted genes. Genes are classified according to the changing pattern during development.

Fig VIII. Contact heatmap near the *KCNQ1* imprinted gene during IVF and PA embryogenesis. Quantitatively ISSs are plotted in the middle panel.

Second round of review

Reviewer 1

Li et al have answered my questions to satisfaction. However, some the requested figures have ended up only in the rebuttal and not in the manuscript. I would suggest adding them as a supplementary figure.

Also I asked for a scatterplot of the compartment scores, but the authors provided scatterplots of the compartment strengths. Please create a scatterplot of the compartment scores (i.e. the 1st eigen vector of the observed/expected matrix) and add this to the manuscript so that readers can compare the compartmentalization.

Reviewer 2

The authors have address most, but not all the concerns that I initially had about these data.

In particular, and in agreement with point #17 from reviewer #1, the data in Figure 6 still look very unconvincing to me. This is particularly highlighted with the zoom in plots provided in Fig VIII in the rebuttal (please note that these are not provided in the supplementary material). I fail to see any particularly noticeable structure in these data, which calls into question the analysis of insulation score changes in these regions (Fig 6A, B), since I don't think that the results are meaningful with the current level of resolution and quality of the data. So I do not think that this section is supported with the data presented in the manuscript.

In addition, I can't find information regarding how the heatmaps in Fig 4SA and Fig S8A were generated. These should be included in the methods. In particular, is the TAD order the same in the three set of plots? Looking at these plots and assuming that the TAD boundary order would be similar between the plots, to me these results suggest rather than a general individual asynchrony in the establishment of insulation, a developmental delay in these samples.

Reviewer 1

Li et al have answered my questions to satisfaction. However, some the requested figures have ended up only in the rebuttal and not in the manuscript. I would suggest adding them as a supplementary figure.

Answer: Thank you for the suggestion. We have added all figures in the rebuttal to the supplementary.

Also I asked for a scatterplot of the compartment scores, but the authors provided scatterplots of the compartment strengths. Please create a scatterplot of the compartment scores (i.e. the 1st eigen vector of the observed/expected matrix) and add this to the manuscript so that readers can compare the compartmentalization.

Answer: Thank you for the suggestion. We created the scatterplot of the compartment score and added it as Fig S7B (shown below). We can see that the compartments assignment was similar between PA and AG embryos and the 4 cell stage display the smallest correlation.

Fig S7B. Scatter plots of the compartments scores in chromosome 14. Each dot represents the PC1 value of a bin (binsize=300kb).

Reviewer #2: The authors have address most, but not all the concerns that I initially had about these data.

In particular, and in agreement with point #17 from reviewer #1, the data in Figure 6 still look very unconvincing to me. This is particularly highlighted with the zoom in plots provided in Fig VIII in the rebuttal (please note that these are not provided in the supplementary material). I fail to see any particularly noticeable structure in these data, which calls into question the analysis of insulation score changes in these regions (Fig 6A, B), since I don't think that the results are meaningful with the current level of resolution and quality of the data. So I do not think that this section is supported with the data presented in the manuscript.

Answer: Thank you for the suggestion. We agreed with the reviewer that the resolution of current data may not sufficient for analysis of single gene loci and this is the problem for all sisHi-C analysis using low-input material. We delete the original Fig. 6A and 6B according to the suggestion. As the GO analysis was applied on all the genes in the changed TAD boundaries, we think the enrichment of certain terms shall be informative, and we moved the Fig 6C into Supplementary Figure S11B.

In addition, I can't find information regarding how the heatmaps in Fig 4SA and Fig S8A were generated. These should be included in the methods. In particular, is the TAD order the same in the three set of plots? Looking at these plots and assuming that the TAD boundary order would be similar between the plots, to me these results suggest rather than a general individual asynchrony in the establishment of insulation, a developmental delay in these samples.

Answer: Thank you for the question. The heatmaps for all samples were ordered by the IS of corresponding borders in PEF. So the TAD order were same for Fig S4A and S8A, and plot shows the IS profile in a 1.25Mb region flanking the TAD borders at 50kb resolution. We have added all the details into the Methods.

Indeed, we agree with the reviewer that the pattern we observed from PA and AG cannot fully represent what happened for the paternal and maternal chromosomes *in vivo*. Although, we observed asynchrony in mouse, e.g, the progress of TAD establishment is faster in mouse maternal chromosomes than those in paternal at zygote, early 2 cell and late 2 cell stages (Fig. 5D), the existence of asynchrony in mouse can only indicate the possibility of asynchrony in pig may not be zero. Based on this fact, we did not intend to claim asynchrony in the general individuals in pig, but described there was different developmental paces between the uniparental embryos, or as pointed out by the reviewer, the delayed development in AG.

Therefore, it would be invaluable if homozygote pig lines could be established in the future. To clarify this fact, we added additional discussion on this issue in the Discussion part.